# EXPLAINING THE REASONING OF LARGE LANGUAGE MODELS USING ATTRIBUTION GRAPHS

## ABSTRACT

Large language models (LLMs) exhibit remarkable capabilities, yet their reasoning remains opaque, raising safety and trust concerns. Attribution methods, which assign credit to input features, have proven effective for explaining the decision making of computer vision models. From these, context attributions have emerged as a promising approach for explaining the behavior of autoregressive LLMs. However, current context attributions produce incomplete explanations by directly relating generated tokens to the prompt, discarding inter-generational influence in the process. To overcome these shortcomings, we introduce the Context Attribution via Graph Explanations (CAGE) framework. CAGE introduces an *attribution graph*: a directed graph that quantifies how each generation is influenced by both the prompt and all prior generations. The graph is constructed to preserve two properties—causality and row stochasticity. The attribution graph allows context attributions to be computed by marginalizing intermediate contributions along paths in the graph. Across multiple models, datasets, metrics, and methods, CAGE improves context attribution faithfulness, achieving average gains of up to $40\%$.

## 1 INTRODUCTION

The rise of autoregressive large language models (LLMs) has led to widespread adoption across domains, from chatbots (Achiam et al., 2023) to autonomous agents (Wang et al., 2024) and code generation (Kirchner & Knoll, 2025). Autoregressive language models generate an output sequence one token at a time, conditioning each token on the entire prefix—the input and all previously generated tokens. As these models continue to scale (Kaplan et al., 2020), so does the opacity of their multi-step decision-making, making interpretability increasingly critical. To address this challenge, a variety of explanation strategies have been explored (Zhao et al., 2024), including self-explanation through training (Camburu et al., 2018), prompting (Bills et al., 2023), counterfactual reasoning (Kamahi & Yaghoobzadeh, 2024), training data analysis (Grosse et al., 2023), mechanistic interpretability (Ameisen et al., 2025), and attribution methods (Cohen-Wang et al., 2024).

Among these techniques, attribution methods have emerged as a promising paradigm. They estimate the contribution of each input feature to an output, and have proven highly effective for computer vision models (Das & Rad, 2020; Dwivedi et al., 2023) and LLMs solving classification problems (Atanasova, 2024; Modarressi et al., 2023; Enguehard, 2023; Barkan et al., 2024a;b). Since each autoregressive generation step is itself a token-level classification problem, any attribution method from the classification literature can be used to explain the generation of that token. However, explaining the output of an autoregressive LLM requires more than a local stepwise attribution. The explanation must satisfy two desiderata: (1) explain any generation with respect to the prompt, and (2) trace causal reasoning from the prompt through prior generations to the generation(s) of interest.

To this end, the current paradigm of context attribution, coined by Cohen-Wang et al. (2024), aims to explain any output of the model (a subset of the generated tokens) with respect to the prompt. To do so, they apply existing *base attribution methods* $\mathcal{M}$ to every generation of the LLM (Liu et al., 2024; Zhao & Shan, 2024). However, they only capture *how each generated token is directly influenced by prompt tokens*, discarding the inter-generational influences in the process. This omission undermines explanation quality, especially in chain-of-thought reasoning (Shojaee et al., 2025; Wei et al., 2022) where intermediate generations provide essential information for later outputs. Moreover, when explaining an output, context attributions aggregate scores by summation, implicitly assuming

that all intermediate tokens contribute equally. These simplifications erase the causal structure of autoregressive generation and produce incomplete and misleading explanations.

In this paper, we propose Context Attribution via Graph Explanations (CAGE), a novel framework for constructing context attribution using an *attribution graph* that overcomes the aforementioned shortcomings and generalizes across base attribution methods. Our contributions are as follows:

1. We propose an *attribution graph* representation for explaining the reasoning of autoregressive LLMs. The graph is constructed using any base attribution method $\mathcal{M}$ that captures how each generated token is influenced by both the prompt and prior generations. This attribution graph is constructed with two properties—causality and row stochasticity—ensuring that LLM reasoning chains are faithfully modeled.

2. We extract a context attribution from the attribution graph by marginalizing the contributions of all prior tokens along causal paths in the graph from any generation(s) to the prompt tokens. The attribution graph itself also visualizes (1) prompt-level explanations and (2) reasoning pathways within the chain-of-thought process.

3. The experimental evaluation shows that our CAGE framework consistently enhances the quality of multiple attribution methods, achieving maximum and average improvements of up to $134\%$ and $40\%$, respectively, across diverse models and datasets.

The remainder of the paper is organized as follows: background in Section 2, the methodology in Section 3, experimental evaluation in Section 4, and the conclusion in Section 5.

## 2 BACKGROUND

In this section, we formulate the problem of attributing an autoregressive LLM, first defining the goals of the attribution, then contrasting existing approaches with these goals.

**Autoregressive Generation.** Consider the input to an autoregressive LLM as two sets: the prompt tokens $\mathcal{P} = \{x_1, \ldots, x_P\}$ and the generated tokens $\mathcal{Y}_{<t} = \{x_{P+1}, \ldots, x_{t-1}\}$. The token $x_t$ is produced by computing a probability distribution over the vocabulary conditioned on the input tokens. This conditional distribution is parameterized by the model weights $\theta$:

$$x_t \sim P(x_t \mid \mathcal{P}, \mathcal{Y}_{<t}; \theta).$$

Each generation step is then a classification over the vocabulary, where the input grows with each $x_t$, until the final token $x_T$ is predicted. The complete token sequence is then $(x_1, \ldots, x_T)$ with $T = P + Y$, where there are $Y$ generated tokens $\mathcal{Y} = \{x_{P+1}, \ldots, x_T\}$.

**Base Attribution.** The attribution of any single-step classification to an input is defined as a base attribution $\mathcal{M}$, which serves as the primary unit for LLM attribution. Formally, for an autoregressive LLM $f_\theta$, a base attribution is the mapping:

$$\mathcal{M} : (f_\theta, x_t, (x_1, ..., x_{t-1})) \mapsto \boldsymbol{s}^{x_t} = (s_1, \ldots, s_{t-1}),$$

where $\boldsymbol{s}^{x_t}$ is a vector of attribution scores over the input tokens. Each score $s_i$ captures the direct effect of $x_i \in \{x_1, \ldots, x_{t-1}\}$ on $x_t$. $\mathcal{M}$ can be any attribution method developed for classification.

### 2.1 PROBLEM STATEMENT

A context attribution for an autoregressive LLM (Cohen-Wang et al., 2024) explains a portion of the generation, an output $\mathcal{O} \subseteq \mathcal{Y}$, by identifying how prompt tokens $\mathcal{P}$ causally contribute to the generated outputs. Formally a context attribution is the mapping:

$$\mathcal{M}_{\text{CA}} : (f_\theta, \mathcal{O}, \mathcal{P}) \mapsto \boldsymbol{a}^O = (\alpha_1, \ldots, \alpha_P).$$

Unlike base attributions, which explain a single prediction with respect to all preceding inputs, context attributions (i) evaluate a set of generated tokens $\mathcal{O}$ and (ii) restrict attribution to the prompt $\mathcal{P}$.

The challenge is how to adapt base attribution methods $\mathcal{M}$, originally designed for single-step classification, to autoregressive generation in a way that (1) explains outputs with respect to the prompt, and (2) traces causal influence through intermediate generations. To reduce cost, prior work explains at the sentence level (Liu et al., 2024); we follow this practice while presenting token-level definitions without loss of generality.

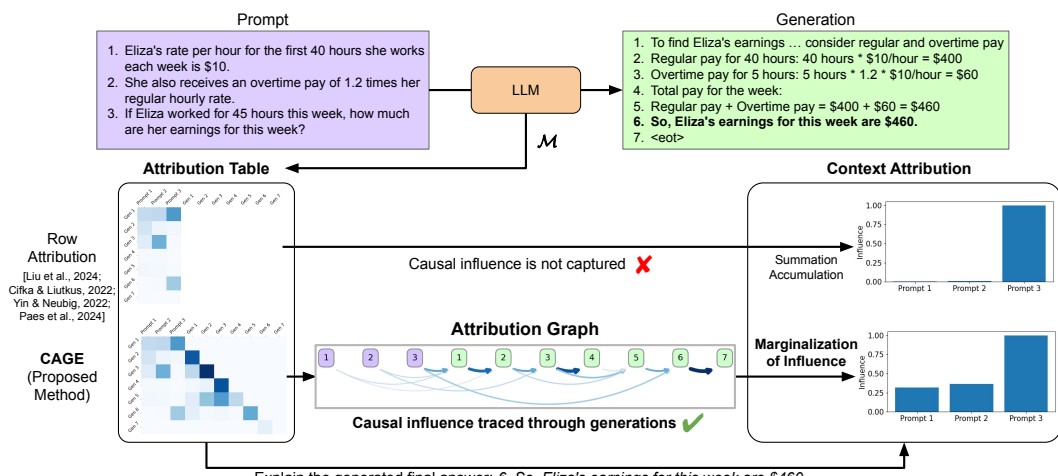

Figure 1: Context attributions explain an autoregressive LLM by identifying how prompt tokens causally influence its output. Current row attribution approaches (middle row) apply a base attribution method $\mathcal{M}$ at each generation step, summing only direct prompt influence and discarding inter-generational effects, thus missing causal reasoning. CAGE (bottom row) instead constructs an attribution graph that captures both prompt and inter-generational influence, then marginalizes influence along its paths to produce faithful, causality-respecting context attributions.

## 2.2 RELATED WORK

Several methods have been proposed for computing context attributions. These methods either fall into the category of surrogate models (Cohen-Wang et al., 2024; Paes et al., 2024; Cohen-Wang et al., 2025) or attribution methods (Kamahi & Yaghoobzadeh, 2024; Yin & Neubig, 2022; Cífka & Liutkus, 2022; Zhao & Shan, 2024; Liu et al., 2024; Li et al., 2024; Vafa et al., 2021), which we denote as *row attributions* and are illustrated in the middle row of Figure 1. We call them row attributions because they all use a similar row-based methodology. We do not further review or compare with surrogate models, as they have their own advantages and limitations.

**Row Attributions** compute a context attribution by applying a base attribution $\mathcal{M}$ to each token in $\mathcal{O}$. This yields a set of influence vectors, $\{\boldsymbol{s}^{x_t} \mid x_t \in \mathcal{O}\}$. Next, the attributions to generated tokens are zeroed out and the result is stored in an attribution table, which is shown in the middle-left of Figure 1. The final context attribution is then computed by summing the resulting vectors, $\boldsymbol{a}^{\mathcal{O}} = \sum_{x_t \in \mathcal{O}} \boldsymbol{s}^{x_t}$, thereby assuming all intermediate generations contribute equally. Row attribution was first introduced with input perturbation base methods (Zeiler & Fergus, 2014; Fisher et al., 2019). These studies have focused on enhancing context attribution by improving the base method in multiple ways: adjusting the order and scale of perturbations (Liu et al., 2024; Vafa et al., 2021), how the change in model output is measured over the generation (Cífka & Liutkus, 2022; Li et al., 2024), and what perturbation baselines are used for token replacement (Zhao & Shan, 2024). These works have also utilized a number of sensitivity-based methods such as gradients (Simonyan et al., 2013) and integrated gradients (Sundararajan et al., 2017) as well as attention-based methods (Vaswani et al., 2017; Abnar & Zuidema, 2020). These base methods were converted to row attributions for reference comparisons (Yin & Neubig, 2022; Kamahi & Yaghoobzadeh, 2024).

We observe that all row attributions ignore the LLM's causal reasoning, misrepresent desideratum (1), fail to satisfy desideratum (2), and often produce incomplete or misleading explanations in settings such as chain-of-thought reasoning. We see in Figure 1 that a base method $\mathcal{M}$ provides an attribution table of token-to-token influences, but row attribution (middle row) discards inter-generational effects and simply sums the direct influences in the selected rows. As a result, they do not capture causal relationships (desideratum (2) unsatisfiable) and do not attribute critical prompt tokens necessary for answering the question (desideratum (1) not properly met). To overcome these limitations, we propose a new representation called an *attribution graph* that captures the causal influence between all tokens. The graph enables context attributions to be computed by marginalization of intermediate generations, which is shown at the bottom of Figure 1.

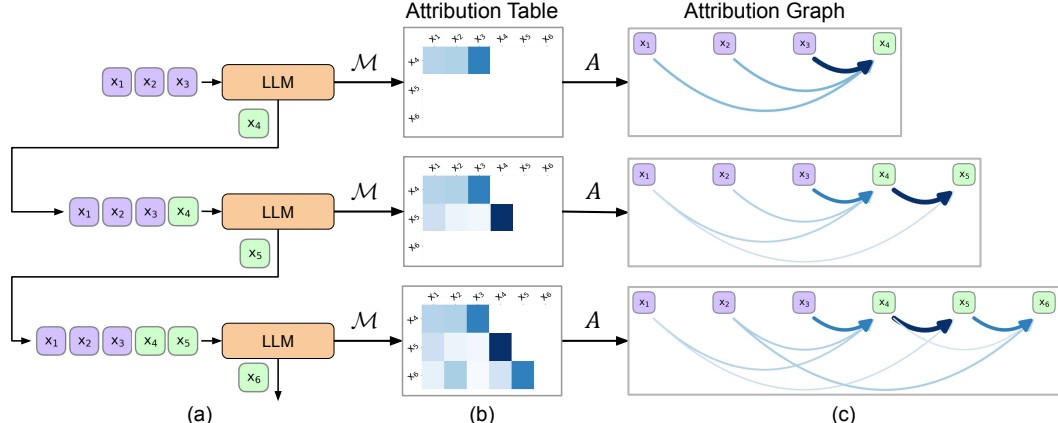

Figure 2: We illustrate the construction of the attribution graph. At each LLM generation step (a), we apply a base attribution $\mathcal{M}$, to measure the influence of the current input on the generation. We perform a nonnegative normalization of the influence values and add them to the adjacency matrix (b) of the attribution graph (c) that captures the causal influence of the generation process.

## 3 METHODOLOGY

In this section, we propose the CAGE framework for computing context attributions of autoregressive LLMs, consisting of three components. First, we introduce the definition of our attribution graph that captures how causal influence propagates from the prompt through each generated token. Second, we describe how the attribution graph is constructed using any base attribution method $\mathcal{M}$. Then, we compute a context attribution for any output token(s), by marginalizing the contributions of intermediate tokens through a path in the graph from the output to the prompt. Finally, we provide a discussion of the motivation for the design choices of this framework to ground.

### 3.1 THE ATTRIBUTION GRAPH

We define the attribution graph $\mathcal{G} = (V, E, w)$, where the vertices $V = \mathcal{P} \cup \mathcal{Y} = \{v_1, \ldots, v_T\}$ consist of both the prompt and generated tokens, and the edges $E \subseteq V \times V$ have weights $w(u, v)$ which capture prediction influence from prior tokens to future generated tokens. It has two properties:

1. **Causality:** Edges point forward in time, from earlier tokens to later generated tokens
$$E \subseteq \{(v_i, v_j) \mid (i < j \leq T) \wedge (P < j)\}.$$

2. **Row Stochastic:** For $v_{t>P}$, incoming edge weights are nonnegative and sum to 1
$$\sum_{i=1}^{t-1} w(v_i, v_t) = 1, \quad w(v_i, v_t) \geq 0.$$

This is, by definition, a directed acyclic graph (DAG). Due to the causality constraint, the sources of the graph are the prompt tokens and the sink of the graph is $v_T$ (e.g., EOS token). Later, we allow any $v_\tau$, $P < \tau \leq T$, to serve as a sink for building the context attribution.

These constraints enable two key features of the attribution graph that support its use for LLM explanation. *Sparsity and interpretability:* a pruning value on the range $[0, 1]$ can be enforced over the graph to remove low-influence edges, reducing visual clutter, and promoting human interpretable tracing of the model's causal reasoning (Miller, 1956). *Context attribution:* the influence of the prompt tokens on any generated token can be computed by marginalizing the influences of all prior generations, producing a context attribution which captures causal influence (see Section 3.3).

### 3.2 CONSTRUCTING THE ATTRIBUTION GRAPH

In this section, we explain how to construct an attribution graph $\mathcal{G}$ with respect to a prompt $\mathcal{P}$ and generation $\mathcal{Y}$, which is illustrated in Figure 2. The generation of $\mathcal{Y}$ from the prompt $\mathcal{P}$ by an LLM

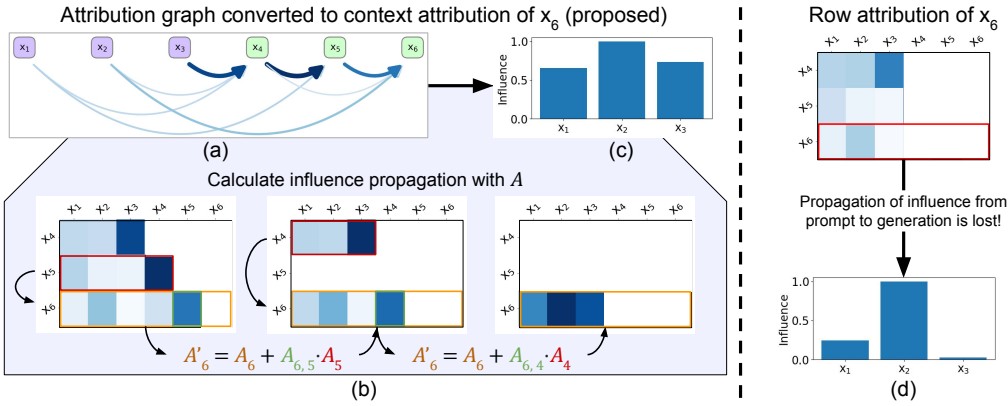

Figure 3: An illustration of how CAGE converts an attribution graph (a) to a context attribution for the generation $x_6$ (c). We iteratively propagate influence, as defined by Eq. 2, to incorporate the causal influence from previous tokens into the explained generation (b), producing our context attribution (c). In contrast, current context attributions (d) do not capture influence propagation.

is shown in Figure 2(a). We collect the outputs of a base attribution $\mathcal{M}$ across all generations $\mathcal{Y}$. Recall, for each token $x_t$, $t > P$, $\mathcal{M}$ produces a score vector $s^{x_t}$ capturing the direct influence of the preceding tokens, which is shown in Figure 2(b). By capturing these vectors for all generations, we create an attribution table $\boldsymbol{T} \in \mathbb{R}^{Y \times T}$ such that each row is the raw attribution scores of a generation, $\boldsymbol{T}_{t-P,:} = s^{x_t}$, and each value $\boldsymbol{T}_{i,j}$ is the raw influence of token $j$ on token $i$. $\boldsymbol{T}$ is naturally separated into two regions: (1) the first $P$ columns, which capture direct prompt-to-generation influence, and (2) columns $P + 1$ to $T$, which form a strictly lower-diagonal block that captures inter-generational influence. As a reminder, existing context attribution methods populate only the first region, leaving the lower-diagonal structure unmodeled (Zhao & Shan, 2024; Cífka & Liutkus, 2022).

We then construct the adjacency matrix $\boldsymbol{A}$ of $\mathcal{G}$, by normalizing the values of $\boldsymbol{T}$ to be row stochastic:

$$\boldsymbol{A}_{i,j} = \frac{\Phi(\boldsymbol{T}_{i,j})}{\sum_j \Phi(\boldsymbol{T}_{i,j})}, \quad i \in \{1, \ldots, Y\}, \quad j \in \{1, \ldots, T\}, \tag{1}$$

where $\Phi(x) = \max(x, 0)$ ensures nonnegativity. Each entry $\boldsymbol{A}_{i,j}$ then represents the weight of the directed edge from node $j$ to node $i$, $w(v_j, v_i)$ as seen in Figure 2(c).

## 3.3 FROM ATTRIBUTION GRAPH TO CONTEXT ATTRIBUTION

Now, given the attribution graph, we wish to compute a context attribution for any output $\mathcal{O} \subseteq \mathcal{Y}$. To do so, we need to capture the total influence of the prompt tokens $\mathcal{P}$ on $\mathcal{O}$. Let us first consider $\mathcal{O} = \{x_\tau\}$, $P < \tau \le T$, where we are explaining one generated token with respect to the prompt. From the attribution graph view, we wish to set this token as the sink of the graph, and propagate all influence from the prompt tokens to the sink.

Consider the adjacency matrix $\boldsymbol{A} \in \mathbb{R}^{Y \times T}$. It captures how the token $x_\tau$ is directly influenced by all tokens $\mathcal{P} \cup \mathcal{Y}_{<\tau}$, where the prior generation influences serve as coefficients for the propagation of prompt influence. It is implicit that the prompt tokens $\mathcal{P}$ have no self influence, so for mathematical clarity, let us consider $\boldsymbol{A} \in \mathbb{R}^{T \times T}$ where the first $P$ rows are all zero, and the matrix is strictly lower diagonal. Then, one step of propagating influence from $x_{\tau-1}$ to $x_\tau$ is formulated as:

$$\boldsymbol{A}'_{\tau,:} = \boldsymbol{A}_{\tau,:} + \boldsymbol{A}_{\tau,\tau-1} \cdot \boldsymbol{A}_{\tau-1,:}. \tag{2}$$

Then, to calculate the context attribution $\boldsymbol{a}^{x_\tau}$, we expand this single step into the sum:

$$\boldsymbol{a}^{x_\tau} = \boldsymbol{A}_{\tau,:} + \sum_{i=\tau-1}^{1} \boldsymbol{A}_{\tau,i} \cdot \boldsymbol{A}_{i,:}. \tag{3}$$

The indices $i = \{1, ..., P\}$ of this vector are the reported values of the context attribution (attribution to the prompt tokens). As a result of this construction, the vector $\boldsymbol{a}^{x_\tau}_{\{1,...,P\}}$ sums to 1, maintaining

its row stochasticity. Figure 3 shows an example of creating $\boldsymbol{a}^{x_6}$ (with $\boldsymbol{A} \in \mathbb{R}^{Y \times T}$). From the attribution graph (a), we propagate influence according to this equation (b), producing the attribution (c). In contrast, existing methods for context attribution fail to explain $x_6$ because they ignore this propagation of influence, misallocating attribution across the prompt (d).

This calculation can be described as solving for one row $\tau$ of a total influence matrix $\boldsymbol{Y} \in \mathbb{R}^{T \times T}$. This matrix captures, for every token, the propagation of influence of the prompt tokens through all prior tokens. Therefore, we can extend this calculation to solve for the full matrix $\boldsymbol{Y}$ as:

$$\boldsymbol{Y} = \boldsymbol{A}(\boldsymbol{I} - \boldsymbol{A})^{-1}, \tag{4}$$

where $\boldsymbol{I} \in \mathbb{R}^{T \times T}$ is the identity matrix. Now, to capture the total influence of the prompt $\mathcal{P}$ on any output $\mathcal{O} \subseteq \mathcal{Y}$, i.e. compute the context attribution $\boldsymbol{a}^{\mathcal{O}}$, we simply sum all rows of $\boldsymbol{Y}$ that correspond to the indices of the output $\mathcal{I}(\mathcal{O})$:

$$\boldsymbol{a}^{\mathcal{O}} = \sum_{\tau \in \mathcal{I}(\mathcal{O})} \boldsymbol{Y}_{\tau,:}, \tag{5}$$

where we report the values $\boldsymbol{a}^{\mathcal{O}}_{\{1,\ldots,P\}}$. As we can see, once we have acquired $\boldsymbol{A}$, we can efficiently create a context attribution for any selected output.

### 3.4 DISCUSSION OF MODELING ASSUMPTIONS

The CAGE framework is designed as a structured abstraction for tracing influence propagation through autoregressive generations, rather than a direct interpretation of the model's internal computation. Below, we position the key modeling assumptions within prior interpretability practice.

**Non-negativity and row-stochasticity** Our construction enforces that all incoming influences to a generated token are non-negative and sum to one. These properties ensure that the attribution graph defines a stable and interpretable flow of influence between tokens. While negative attribution values can represent valuable inhibitory effects, propagating these values through the graph will produce unstable magnitudes, oscillating signs, and result in information cancellation. Similar challenges have led several explanation frameworks to suppress intermediate negative signals, including Layer-wise Relevance Propagation (LRP) (Binder et al., 2016) and Guided Backpropagation (Springenberg et al., 2014) to improve faithfulness and interpretability. Furthermore, the row stochasticity property provides a probabilistic normalization aligned with attention rollout (Abnar & Zuidema, 2020), which assumes conservation of information across layers to enable cumulative analysis of multi-step interactions. Together, these properties allow the graph to support closed-form solution, building a stable and interpretable context attribution as a result. See Appendix A.5 for an ablation study.

**Linear influence propagation** Equations 2 through 4 inherently adopt a linear model of how influence propagates through each generation. We do not intend to approximate the full nonlinearity of the transformer architecture. Rather, this assumption reflects common practices in interpretability research, including local linear surrogate models (Ribeiro et al., 2016), first-order/Taylor methods such as Guided Backpropagation (Springenberg et al., 2014) and Integrated Gradients (Sundararajan et al., 2017), attention-flow and rollout methods (Abnar & Zuidema, 2020), and relevance-propagation frameworks (Binder et al., 2016). These approaches employ linear relaxations to expose the dominant factors that drive model behavior. In our setting, this linear propagation enables CAGE to capture influence across generations, fixing a critical failure of existing row attribution approaches, while remaining compatible with a wide range of base attributions.

## 4 EVALUATION

We evaluate CAGE with quantitative and qualitative analyses. Across two variants of Llama 3 (3B, 8B) (Grattafiori et al., 2024) and Qwen 3 (4B, 8B) (Yang et al., 2025), we compare five row attribution methods against CAGE. Since CAGE focuses on capturing the inter-generational dependencies between output tokens, explaining a generation which is only one token (or sentence) would be a degenerate case where it is equivalent to row attribution. Therefore, to properly evaluate CAGE, we employ three datasets which require the model to use both multi-step information retrieval and chain-of-thought reasoning. To assess performance under these settings, we evaluate using both ground truth coverage and attribution faithfulness metrics. Furthermore, in Appendix A.5 we provide an ablation study of CAGE to verify the need for its non-negative, row stochastic construction.

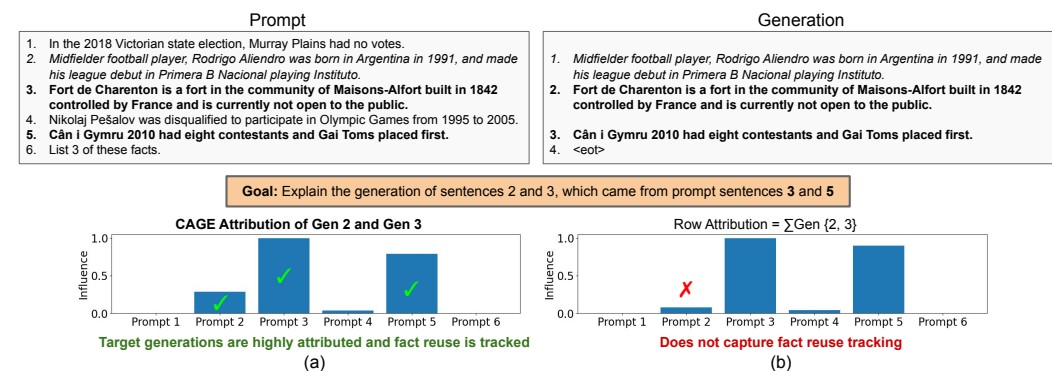

Figure 4: In this Facts dataset example, the output to explain is the second and third generations (prompts 3 and 5). Both CAGE (a) and the row attribution (b) attribute these prompt sentences, but only CAGE also attributes prompt 2, showing how the model tracks prior generations to avoid reuse. Row attribution does not capture this inter-generational influence.

## 4.1 ATTRIBUTION METHODS

We compare against row attributions by mapping their base methods to CAGE's graph as described by the methodology. The first three methods are explicitly defined as row attributions: **perturbation (Pert.)** (Liu et al., 2024) replaces individual sentences with EOS tokens and measures the resulting drop in log-probability of the generated tokens. **Context Length Probing (CLP)** (Cífka & Liutkus, 2022) perturbs in the same manner but measures the Kullback-Leibler divergence over the vocabulary, capturing shifts in the prediction distribution. **ReAGent** (Zhao & Shan, 2024) performs input replacements using predictions from a RoBERTa model to avoid out-of-distribution inputs, and we employ the Longformer for larger perturbations (Beltagy et al., 2020). In addition, we adopt several other attribution techniques that have been adapted into row attributions: **Integrated gradients (IG)** uses a 20-step interpolation from a baseline EOS embedding to the input embedding, computing gradient $\times$ input at each step (Sundararajan et al., 2017; Vafa et al., 2021). **Attention-based attribution** collects attention weights for a generated token by summing across all heads and layers (Vaswani et al., 2017; Liu et al., 2024), and we evaluate with Attn $\times$ IG, which multiplies attention by IG scores to produce more discriminative explanations (Chen et al., 2022; Yuan et al., 2021). In Appendix A.1, we explain how token-level attributions are generalized to sentence-level.

## 4.2 DATASETS

**Facts** is constructed from Feverous (Aly et al., 2021) (87,026 verified Wikipedia claims), to create an *information retrieval* task. Each dataset item is a prompt of $N$ sampled claims and the statement: "List $M$ of these facts." paired with a target generation of $M$ sampled prompt facts. $K$ of these "generated" facts are set as the output $\mathcal{O}$ to explanation and we track the prompt indices of the generated facts. In the quantitative evaluation, we employ the dataset with $N = 9$, $M = 3$, $K = 2$. We provide a qualitative analysis over an example from this dateset in the following section.

**Math** follows Kojima et al. (2022) and is a combination of two datasets of multi-step math questions (MultiArith (Roy & Roth, 2016) + SVAMP (Patel et al., 2021)) which require *chain-of-thought reasoning* for answering. From these datasets, we keep all prompts with at least three sentences (1,258 examples). We additionally augment the data with distractor sentences ("Unrelated sentence.") inserted among the prompt sentences. The model responds free-form and the final generated sentence before the EOS is the final answer which we mark for explanation. As such, we track the indices of the original prompt sentences to serve as the ground truth for attribution. The next section will illustrate an example of this dataset with a qualitative analysis.

**MorehopQA** (Schnitzler et al., 2024) is a long-context question-answering dataset which is designed to require *information extraction and chain-of-thought reasoning*. Below is a shortened example

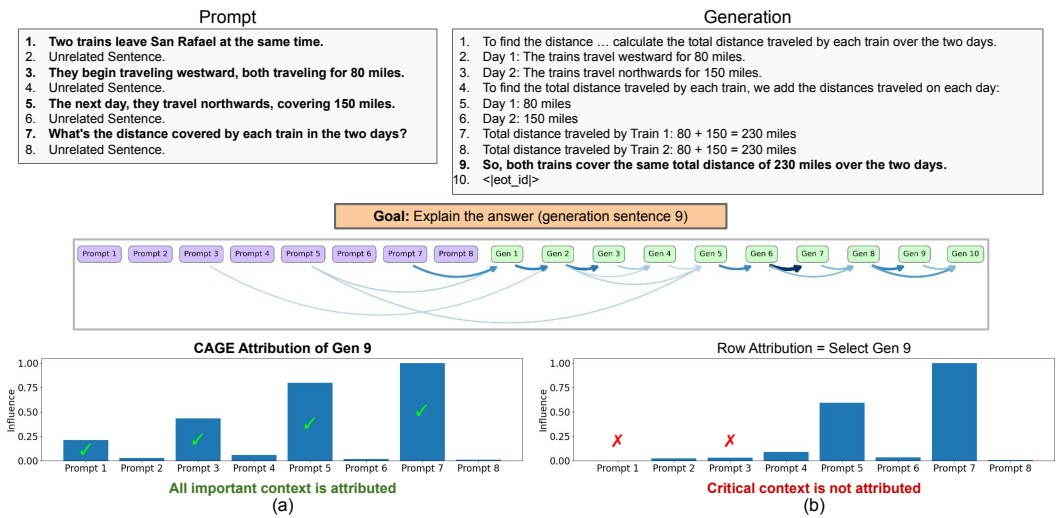

Figure 5: We show an example of the Math dataset where the answer (generated sentence 9) should be explained by attributing all odd prompt sentences. CAGE (a) successfully attributes only and all of these prompt regions because it captures causal influence between generations. The row attribution (b) only attributes a subset of the important prompt sentences, ignoring vital context.

from this dataset. It contains 60 of the original 146 words since the distractor text has been removed.

> The Prince and Me is a 2004 romantic comedy film directed by Martha Coolidge ... Kam Heskin ... is an American actress ... Heskin went to play Paige Morgan in the "The Prince and Me" ... How many letters are there between the first and last letters of the first name of the director of a 2004 film where Kam Heskin plays Paige Morgan in? *Answer: 4 letters ("arth" in Martha)*

### 4.3 QUALITATIVE ANALYSIS ON FACTS AND MATH EXAMPLES

For the Facts dataset, Figure 4 illustrates a Pert. example for $N = 5$, $M = 3$, $K = 2$, where the generations have prompt indices $\{2, 3, 5\}$ and the output to explain is generation indices $\{2, 3\}$. We see both CAGE (a) and row attribution (b) place a majority of attribution on the explained output sentences. However, CAGE displays an additional desirable behavior: it attributes prompt 2, capturing how the model tracks the sentences it already generated to prevent reuse. Because prompt 3's reuse is also tracked, it receives more attribution than prompt 5. In contrast, row attribution does not show this behavior, illustrating its failure to capture inter-generational influence and therefore model reasoning.

For the Math dataset, Figure 5 illustrates a Pert. example for explaining the answer, which is generation index 9. Because CAGE captures inter-token influence across all generations, the resulting context attribution (a) properly attributes every important prompt sentence. On the other hand, the row attribution (b) fails to properly attribute prompt indices $\{1, 3\}$ which contain critical context for answering the math question. In the next section, we evaluate this behavior quantitatively.

In Appendix A.4, we provide six more qualitative examples for all three datasets for both Attn $\times$ IG and perturbation. We find the same behavior seen here extents to these additional results.

### 4.4 COT REASONING GROUND TRUTH EVALUATION

We quantitatively evaluate how well a context attribution identifies critical prompt sentences in the Math questions using an attribution coverage (AC) metric. Since distractor sentences should not receive attribution, high-quality explanations concentrate attribution on the important prompt sentences, yielding a high AC. At the same time, attribution should be approximately uniformly distributed across ground-truth sentences so that none are missed. Formally, we define AC as:

$$\mathrm{AC}(\boldsymbol{a}) = \frac{1}{|GT|} \sum_{j \in GT} \mathbf{1}\left(\frac{1}{2}\mathbb{E}(\boldsymbol{a}_{GT}) \leq r_j \leq \frac{3}{2}\mathbb{E}(\boldsymbol{a}_{GT})\right), \quad r_j = \frac{\boldsymbol{a}_j}{\sum_i \boldsymbol{a}_i},$$

Table 1: Attribution coverage evaluation on the Math dataset

| Dataset | | Math | | | |
|---|---|---|---|---|---|
| Model | | Llama 3B | Llama 8B | Qwen 4B | Qwen 8B |
| Metric ($\uparrow$) | | AC | AC | AC | AC |
| IG | Row | $0.551 \pm 0.28$ | $\mathbf{0.160} \pm 0.23$ | $0.382 \pm 0.29$ | $\mathbf{0.582} \pm 0.27$ |
| | CAGE (ours) | $\mathbf{0.635} \pm 0.28$ | $0.151 \pm 0.24$ | $\mathbf{0.414} \pm 0.29$ | $0.580 \pm 0.27$ |
| Attn $\times$ IG | Row | $0.518 \pm 0.27$ | $0.148 \pm 0.22$ | $0.353 \pm 0.30$ | $\mathbf{0.406} \pm 0.26$ |
| | CAGE (ours) | $\mathbf{0.621} \pm 0.27$ | $\mathbf{0.153} \pm 0.24$ | $\mathbf{0.378} \pm 0.29$ | $0.399 \pm 0.24$ |
| CLP | Row | $0.411 \pm 0.27$ | $0.441 \pm 0.29$ | $0.368 \pm 0.25$ | $0.423 \pm 0.28$ |
| | CAGE (ours) | $\mathbf{0.483} \pm 0.28$ | $\mathbf{0.502} \pm 0.30$ | $\mathbf{0.456} \pm 0.28$ | $\mathbf{0.446} \pm 0.28$ |
| ReAGent | Row | $0.252 \pm 0.28$ | $0.309 \pm 0.31$ | $0.196 \pm 0.24$ | $0.250 \pm 0.25$ |
| | CAGE (ours) | $\mathbf{0.437} \pm 0.28$ | $\mathbf{0.472} \pm 0.28$ | $\mathbf{0.458} \pm 0.28$ | $\mathbf{0.436} \pm 0.29$ |
| Pert. | Row | $0.237 \pm 0.24$ | $0.262 \pm 0.26$ | $0.191 \pm 0.23$ | $0.230 \pm 0.24$ |
| | CAGE (ours) | $\mathbf{0.420} \pm 0.26$ | $\mathbf{0.437} \pm 0.28$ | $\mathbf{0.442} \pm 0.28$ | $\mathbf{0.425} \pm 0.28$ |
| Wins | | 5/5 | 4/5 | 5/5 | 3/5 |

where $\boldsymbol{a}$ is a context attribution, $GT$ indexes the ground-truth sentences, $\mathbb{E}(\boldsymbol{a}_{GT}) = \frac{1}{|GT|}$ corresponds to a uniform share of attribution across $GT$, and $r_j$ is the fraction of the total attribution assigned to the $j$-th ground-truth sentence. The AC score therefore measures the fraction of ground-truth sentences whose attribution falls within the expected range, with AC $= 1$ indicating perfect coverage and AC $= 0$ meaning that every ground-truth sentence was missed.

In Table 1, we present attribution coverage results. For the perturbation base methods we use 500 examples, and for IG methods, we use 250 examples. The table reports the mean and standard deviation of scores for the examples. Across the table, CAGE indicates significant improvements in attribution coverage. Particularly for the ReAGent and Pert base methods, it yields large gains in coverage, showing its ability to extract critical information with its representation of causal influence. Overall, CAGE achieves 17/20 wins for an $85\%$ win rate. Where it falls behind, losses are minor. Across all results, CAGE achieves a maximum and average improvement in AC of $134\%$ and $40\%$, respectively. These results corroborate the behavior seen in Figure 5.

## 4.5 FAITHFULNESS EVALUATION

Following prior LLM attribution work (Li et al., 2024; Zhao & Shan, 2024), we evaluate faithfulness using two input perturbation metrics, RISE (Petsiuk et al., 2018) and MAS (Walker et al., 2024). Both metrics assume that removing highly attributed inputs should decrease the probability of generating the original output. We implement sentence-level deletion tests. The multi-step perturbation process iteratively (and cumulatively) ablates prompt sentences (perturbed with end-of-sequence tokens) (Li et al., 2024) in descending order of attribution magnitude and, at each step, the output probability is measured. The test forms a decreasing generation probability vs perturbation percentage curve. A lower area under this perturbation curve indicates higher faithfulness.

In Table 2, we report the faithfulness evaluation over both metrics for the Llama 3 8B and Qwen 3 8B models, using 500 examples from the MorehopQA and Facts datasets. The table reports the mean and standard deviation of scores for the examples. Across all 5 base attributions, our attribution graph approach wins 40/40 faithfulness tests. It achieves a maximum and average improvement over row attribution of $30\%$ and $11\%$, respectively. In Table 3, we provide the results for the faithfulness evaluation on Math using all four models: Llama 3 3B and 8B and Qwen 3 4B and 8B. Again, CAGE wins all 40 evaluations in the table. Across this table, CAGE achieves a maximum and average improvement over row attribution of $37\%$ and $16\%$, respectively. In Appendix A.2, we provide the remaining results of this test for the smaller models on MorehopQA and Facts. These evaluations illustrate the same trend as these table, where our attribution graph approach wins every faithfulness evaluation. These wins in faithfulness indicate that CAGE better captures the causal reasoning of autoregressive LLMs in context attributions.

Table 2: Faithfulness evaluation via input perturbation metrics on MorehopQA and Facts

| Dataset | | MorehopQA | | | | Facts | | | |
|---|---|---|---|---|---|---|---|---|---|
| Model | | Llama 3 8B | | Qwen 3 8B | | Llama 3 8B | | Qwen 3 8B | |
| Metric (↓) | | RISE | MAS | RISE | MAS | RISE | MAS | RISE | MAS |
| IG | Row | $0.474 \pm 0.22$ | $0.604 \pm 0.19$ | $0.395 \pm 0.15$ | $0.508 \pm 0.17$ | $0.299 \pm 0.10$ | $0.454 \pm 0.16$ | $0.210 \pm 0.08$ | $0.305 \pm 0.10$ |
| | CAGE (ours) | $\mathbf{0.412} \pm 0.19$ | $\mathbf{0.534} \pm 0.16$ | $\mathbf{0.361} \pm 0.13$ | $\mathbf{0.465} \pm 0.15$ | $\mathbf{0.290} \pm 0.11$ | $\mathbf{0.423} \pm 0.16$ | $\mathbf{0.202} \pm 0.08$ | $\mathbf{0.299} \pm 0.09$ |
| Attn × IG | Row | $0.470 \pm 0.21$ | $0.597 \pm 0.19$ | $0.383 \pm 0.15$ | $0.496 \pm 0.18$ | $0.303 \pm 0.10$ | $0.456 \pm 0.15$ | $0.235 \pm 0.09$ | $0.324 \pm 0.11$ |
| | CAGE (ours) | $\mathbf{0.418} \pm 0.20$ | $\mathbf{0.534} \pm 0.15$ | $\mathbf{0.369} \pm 0.14$ | $\mathbf{0.476} \pm 0.16$ | $\mathbf{0.293} \pm 0.11$ | $\mathbf{0.426} \pm 0.16$ | $\mathbf{0.232} \pm 0.09$ | $\mathbf{0.320} \pm 0.11$ |
| CLP | Row | $0.332 \pm 0.11$ | $0.483 \pm 0.16$ | $0.249 \pm 0.10$ | $0.364 \pm 0.16$ | $0.217 \pm 0.07$ | $0.287 \pm 0.10$ | $0.185 \pm 0.10$ | $0.264 \pm 0.19$ |
| | CAGE (ours) | $\mathbf{0.320} \pm 0.11$ | $\mathbf{0.453} \pm 0.15$ | $\mathbf{0.236} \pm 0.10$ | $\mathbf{0.326} \pm 0.14$ | $\mathbf{0.214} \pm 0.07$ | $\mathbf{0.284} \pm 0.10$ | $\mathbf{0.184} \pm 0.10$ | $\mathbf{0.262} \pm 0.19$ |
| ReAGent | Row | $0.358 \pm 0.13$ | $0.505 \pm 0.16$ | $0.307 \pm 0.14$ | $0.435 \pm 0.19$ | $0.234 \pm 0.07$ | $0.350 \pm 0.14$ | $0.209 \pm 0.08$ | $0.305 \pm 0.15$ |
| | CAGE (ours) | $\mathbf{0.319} \pm 0.10$ | $\mathbf{0.439} \pm 0.14$ | $\mathbf{0.233} \pm 0.10$ | $\mathbf{0.315} \pm 0.13$ | $\mathbf{0.220} \pm 0.07$ | $\mathbf{0.318} \pm 0.14$ | $\mathbf{0.160} \pm 0.06$ | $\mathbf{0.215} \pm 0.10$ |
| Pert. | Row | $0.341 \pm 0.11$ | $0.526 \pm 0.16$ | $0.305 \pm 0.14$ | $0.448 \pm 0.20$ | $0.245 \pm 0.08$ | $0.375 \pm 0.15$ | $0.207 \pm 0.08$ | $0.303 \pm 0.14$ |
| | CAGE (ours) | $\mathbf{0.307} \pm 0.10$ | $\mathbf{0.458} \pm 0.15$ | $\mathbf{0.230} \pm 0.09$ | $\mathbf{0.319} \pm 0.13$ | $\mathbf{0.227} \pm 0.08$ | $\mathbf{0.333} \pm 0.15$ | $\mathbf{0.159} \pm 0.06$ | $\mathbf{0.214} \pm 0.10$ |
| Wins | | 5/5 | 5/5 | 5/5 | 5/5 | 5/5 | 5/5 | 5/5 | 5/5 |

Table 3: Faithfulness evaluation via input perturbation metrics on Math

| Dataset | | Math | | | | | | | |
|---|---|---|---|---|---|---|---|---|---|
| Model | | Llama 3 3B | | Llama 3 8B | | Qwen 3 4B | | Qwen 3 8B | |
| Metric (↓) | | RISE | MAS | RISE | MAS | RISE | MAS | RISE | MAS |
| IG | Row | $0.257 \pm 0.08$ | $0.337 \pm 0.15$ | $0.429 \pm 0.11$ | $0.603 \pm 0.15$ | $0.310 \pm 0.14$ | $0.438 \pm 0.19$ | $0.204 \pm 0.06$ | $0.292 \pm 0.06$ |
| | CAGE (ours) | $\mathbf{0.229} \pm 0.06$ | $\mathbf{0.291} \pm 0.08$ | $\mathbf{0.412} \pm 0.13$ | $\mathbf{0.508} \pm 0.14$ | $\mathbf{0.297} \pm 0.12$ | $\mathbf{0.410} \pm 0.17$ | $\mathbf{0.196} \pm 0.05$ | $\mathbf{0.280} \pm 0.05$ |
| Attn × IG | Row | $0.268 \pm 0.08$ | $0.352 \pm 0.12$ | $0.429 \pm 0.11$ | $0.609 \pm 0.16$ | $0.315 \pm 0.13$ | $0.445 \pm 0.19$ | $0.223 \pm 0.06$ | $0.298 \pm 0.08$ |
| | CAGE (ours) | $\mathbf{0.239} \pm 0.06$ | $\mathbf{0.306} \pm 0.08$ | $\mathbf{0.409} \pm 0.13$ | $\mathbf{0.518} \pm 0.15$ | $\mathbf{0.302} \pm 0.12$ | $\mathbf{0.417} \pm 0.17$ | $\mathbf{0.218} \pm 0.06$ | $\mathbf{0.280} \pm 0.07$ |
| CLP | Row | $0.233 \pm 0.07$ | $0.311 \pm 0.12$ | $0.238 \pm 0.06$ | $0.317 \pm 0.11$ | $0.224 \pm 0.11$ | $0.316 \pm 0.18$ | $0.161 \pm 0.05$ | $0.209 \pm 0.08$ |
| | CAGE (ours) | $\mathbf{0.208} \pm 0.05$ | $\mathbf{0.261} \pm 0.08$ | $\mathbf{0.222} \pm 0.04$ | $\mathbf{0.279} \pm 0.07$ | $\mathbf{0.198} \pm 0.08$ | $\mathbf{0.275} \pm 0.13$ | $\mathbf{0.158} \pm 0.04$ | $\mathbf{0.202} \pm 0.06$ |
| ReAGent. | Row | $0.283 \pm 0.11$ | $0.404 \pm 0.17$ | $0.277 \pm 0.10$ | $0.380 \pm 0.15$ | $0.248 \pm 0.12$ | $0.365 \pm 0.19$ | $0.202 \pm 0.10$ | $0.282 \pm 0.16$ |
| | CAGE (ours) | $\mathbf{0.204} \pm 0.05$ | $\mathbf{0.256} \pm 0.07$ | $\mathbf{0.218} \pm 0.04$ | $\mathbf{0.274} \pm 0.06$ | $\mathbf{0.186} \pm 0.08$ | $\mathbf{0.251} \pm 0.11$ | $\mathbf{0.149} \pm 0.04$ | $\mathbf{0.188} \pm 0.06$ |
| Pert. | Row | $0.289 \pm 0.12$ | $0.405 \pm 0.18$ | $0.285 \pm 0.10$ | $0.403 \pm 0.17$ | $0.238 \pm 0.12$ | $0.339 \pm 0.19$ | $0.195 \pm 0.09$ | $0.261 \pm 0.15$ |
| | CAGE (ours) | $\mathbf{0.213} \pm 0.07$ | $\mathbf{0.269} \pm 0.09$ | $\mathbf{0.221} \pm 0.05$ | $\mathbf{0.283} \pm 0.08$ | $\mathbf{0.184} \pm 0.07$ | $\mathbf{0.242} \pm 0.10$ | $\mathbf{0.154} \pm 0.04$ | $\mathbf{0.190} \pm 0.04$ |
| Wins | | 5/5 | 5/5 | 5/5 | 5/5 | 5/5 | 5/5 | 5/5 | 5/5 |

## 5 CONCLUSION AND FUTURE WORK

**Conclusion** This work has advanced context attribution for autoregressive LLMs with the introduction of CAGE: Context Attribution via Graph Explanations, a novel framework that captures LLM reasoning with an attribution graph. The core novelty of CAGE lies in its problem formulation. Existing attribution approaches assume that each generated token depends only on the prompt and as a result they only capture the direct influence of prompt tokens on generated tokens and combine influence across generations with equal weight. We identify this as a fundamental gap in the literature. With our novel attribution graph abstraction, we develop an approach for influence propagation that produces a new kind of explanation: one that respects causality, exposes how intermediate generations contribute to later ones, and provides a visualization that allows reasoning chains to be traced. This leads to a measurable impact as CAGE consistently produces more faithful and more complete explanations than row-based attribution across models, datasets, and base attribution methods.

**Future Work** The assumptions employed in this work provide a coherent and empirically effective foundation for CAGE, but future improvement is possible. As shown in our ablations, relaxing non-negativity or row stochasticity substantially degrades faithfulness and interpretability, yet future work could explore structured signed influence graphs, alternative normalization schemes, or higher-order propagation rules that more closely model the transformer architecture. By design, the CAGE framework is modular and any improvements to base attributions and normalization or propagation rules can be incorporated without modifying the overall framework. We theorize that our CAGE framework will promote new avenues for understanding autoregressive LLM reasoning, leading to an improved ability to debug or change the reasoning processes of these models.

ETHICS STATEMENT

This work does not introduce and new models or data that could pose ethical risks. We propose an algorithm for explaining existing LLMs via input attribution. Consistent with the goals of explainable AI, the method exposes model behavior with the aim to improve transparency and understanding. As such, it does not introduce risks beyond those inherent to using existing models.

REPRODUCIBILITY STATEMENT

Upon acceptance, all code for implementation and evaluation of this work (including datasets) will be publicly released. For this submission, these files will be available as supplementary material. In addition, the paper itself provides sufficient information for replicating the methods and evaluations contained within.

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

# A APPENDIX

This Appendix provides supplementary information and the extended experimental results which are outlined in the main paper. First, we outline the process for converting token-level attributions to the sentence-level attributions that we evaluate in this work. Then, we provide the remaining faithfulness results on these smaller models for both MorehopQA and Facts, while also providing faithfulness evaluations for all models on the Math dataset. Following this, we acknowledge our use of an LLM in this work for grammar checking. Finally, we provide a selection of additional qualitative evaluations to show the consistent visual improvements of our CAGE framework.

## A.1 GENERALIZING TOKEN-LEVEL ATTRIBUTIONS TO SENTENCE-LEVEL

In this work we present context attributions at the sentence level, following prior literature (Cohen-Wang et al., 2024; Liu et al., 2024). Token-level attributions are aggregated into sentence-level scores using the attribution table $T$ (Section 3.2) produced after a full LLM generation:

1. A base method $\mathcal{M}$ is used to attribute each generated token to all previous tokens, yielding the token-to-token attribution matrix $T$.

2. The prompt and generated text is segmented into sentences by splitting on newline characters and then applying the NLTK English sentence tokenizer (Bird et al., 2009).

3. Each token is assigned to its containing sentence.

4. For every sentence pair $(i, j)$, we average the token-to-token attributions over the 2D block of $T$ corresponding to tokens in sentence $i$ influencing tokens in sentence $j$. This produces a sentence-to-sentence attribution matrix.

This general procedure applies to any base method $\mathcal{M}$. However, perturbation-based methods can compute sentence attributions directly by perturbing identified sentences rather than individual tokens (Liu et al., 2024). For perturbation, we follow this direct approach for improved efficiency.

## A.2 EXTENDED FAITHFULNESS EVALUATION

In this section, we present the remaining faithfulness evaluation results that were outlined in the paper. We use 500 examples from all three datasets, MorehopQA, Facts, and Math. We present both the mean and standard deviation of evaluation scores across the examples. The Facts dataset is employed with $N = 9$, $M = 3$, $K = 2$ as described in the main body of the paper.

First, in Table 4, we provide the results for the faithfulness evaluation on MorehopQA and Facts using the smaller Llama 3 3B and Qwen 3 4B models. Once again, we see that CAGE wins all 40 evaluations in the table, illustrating how integrating causal influence into the context attribution improves its faithfulness. Across this table, CAGE achieves a maximum and average improvement over row attribution of 28% and 11%, respectively, following the results from the larger models.

Table 4: Faithfulness evaluation via input perturbation metrics on MorehopQA and Facts

| Dataset | | MorehopQA | | | | Facts | | | |
|---|---|---|---|---|---|---|---|---|---|
| Model | | Llama 3 3B | | Qwen 3 4B | | Llama 3 3B | | Qwen 3 4B | |
| Metric ($\downarrow$) | | RISE | MAS | RISE | MAS | RISE | MAS | RISE | MAS |
| IG | Row | $0.274 \pm 0.09$ | $0.413 \pm 0.15$ | $0.297 \pm 0.15$ | $0.429 \pm 0.19$ | $0.287 \pm 0.10$ | $0.405 \pm 0.16$ | $0.244 \pm 0.12$ | $0.358 \pm 0.15$ |
| | CAGE (ours) | $\mathbf{0.252} \pm 0.09$ | $\mathbf{0.361} \pm 0.13$ | $\mathbf{0.271} \pm 0.13$ | $\mathbf{0.395} \pm 0.16$ | $\mathbf{0.234} \pm 0.09$ | $\mathbf{0.320} \pm 0.13$ | $\mathbf{0.241} \pm 0.12$ | $\mathbf{0.354} \pm 0.14$ |
| Attn $\times$ IG | Row | $0.269 \pm 0.09$ | $0.404 \pm 0.15$ | $0.291 \pm 0.15$ | $0.422 \pm 0.20$ | $0.286 \pm 0.10$ | $0.405 \pm 0.16$ | $0.260 \pm 0.12$ | $0.373 \pm 0.15$ |
| | CAGE (ours) | $\mathbf{0.248} \pm 0.09$ | $\mathbf{0.354} \pm 0.13$ | $\mathbf{0.272} \pm 0.13$ | $\mathbf{0.395} \pm 0.16$ | $\mathbf{0.234} \pm 0.09$ | $\mathbf{0.321} \pm 0.14$ | $\mathbf{0.259} \pm 0.12$ | $\mathbf{0.371} \pm 0.15$ |
| CLP | Row | $0.270 \pm 0.10$ | $0.404 \pm 0.16$ | $0.194 \pm 0.10$ | $0.297 \pm 0.16$ | $0.209 \pm 0.07$ | $0.284 \pm 0.13$ | $0.169 \pm 0.09$ | $0.237 \pm 0.16$ |
| | CAGE (ours) | $\mathbf{0.254} \pm 0.09$ | $\mathbf{0.365} \pm 0.14$ | $\mathbf{0.187} \pm 0.10$ | $\mathbf{0.289} \pm 0.16$ | $\mathbf{0.206} \pm 0.07$ | $\mathbf{0.277} \pm 0.13$ | $\mathbf{0.168} \pm 0.09$ | $\mathbf{0.236} \pm 0.16$ |
| ReAGent | Row | $0.299 \pm 0.13$ | $0.420 \pm 0.16$ | $0.217 \pm 0.13$ | $0.322 \pm 0.19$ | $0.246 \pm 0.09$ | $0.383 \pm 0.18$ | $0.186 \pm 0.10$ | $0.273 \pm 0.18$ |
| | CAGE (ours) | $\mathbf{0.260} \pm 0.10$ | $\mathbf{0.354} \pm 0.13$ | $\mathbf{0.175} \pm 0.10$ | $\mathbf{0.231} \pm 0.14$ | $\mathbf{0.219} \pm 0.09$ | $\mathbf{0.324} \pm 0.18$ | $\mathbf{0.157} \pm 0.08$ | $\mathbf{0.220} \pm 0.15$ |
| Pert. | Row | $0.281 \pm 0.11$ | $0.431 \pm 0.17$ | $0.218 \pm 0.12$ | $0.329 \pm 0.19$ | $0.244 \pm 0.08$ | $0.377 \pm 0.16$ | $0.177 \pm 0.08$ | $0.255 \pm 0.15$ |
| | CAGE (ours) | $\mathbf{0.252} \pm 0.10$ | $\mathbf{0.366} \pm 0.15$ | $\mathbf{0.188} \pm 0.10$ | $\mathbf{0.281} \pm 0.16$ | $\mathbf{0.213} \pm 0.08$ | $\mathbf{0.305} \pm 0.15$ | $\mathbf{0.151} \pm 0.07$ | $\mathbf{0.209} \pm 0.12$ |
| Wins | | 5/5 | 5/5 | 5/5 | 5/5 | 5/5 | 5/5 | 5/5 | 5/5 |

### A.3 LLM USAGE

The authors used a large language model for verifying the grammatical correctness of some passages of this submission. Any changes made due to grammatical feedback from the LLM were thoroughly checked by the authors. The authors take intellectual responsibility for all of the content in this submission and guarantee that the work is original and solely their own.

### A.4 EXTENDED QUALITATIVE ANALYSIS

On the following three pages, we provide six additional qualitative analyses of CAGE against existing row attribution approaches. In order: Figure 6 provides an example from the Facts dataset employing the Attn × IG base method on the Llama 3 8B model. Figure 7 shows a Facts example with the Pert. base method on the Qwen 3 8B model. Figure 8, a Math example with the Pert. base method on the Llama 3 8B model. Figure 9, a Math example with the Attn × IG base method on the Qwen 3 8B model. Figures 10 and 9 show MorehopQA examples with the Pert. base method on the Llama 3 8B and Qwen 3 8B models, respectively. Each figure shows the prompt, generation, the goal of the context attribution (which portion(s) of the generation to explain), the attribution graph, and CAGE and row attribution explanations. Overall, we see CAGE provides improved context attribution.

For the Facts and Math dataset, we annotate the figures in correspondence with the examples in Figures 4 and 5, where we see the same behavior emerges, illustrating CAGE's improved performance. For the MorehopQA dataset, the figure captions describe the desirable behaviors seen in the example.

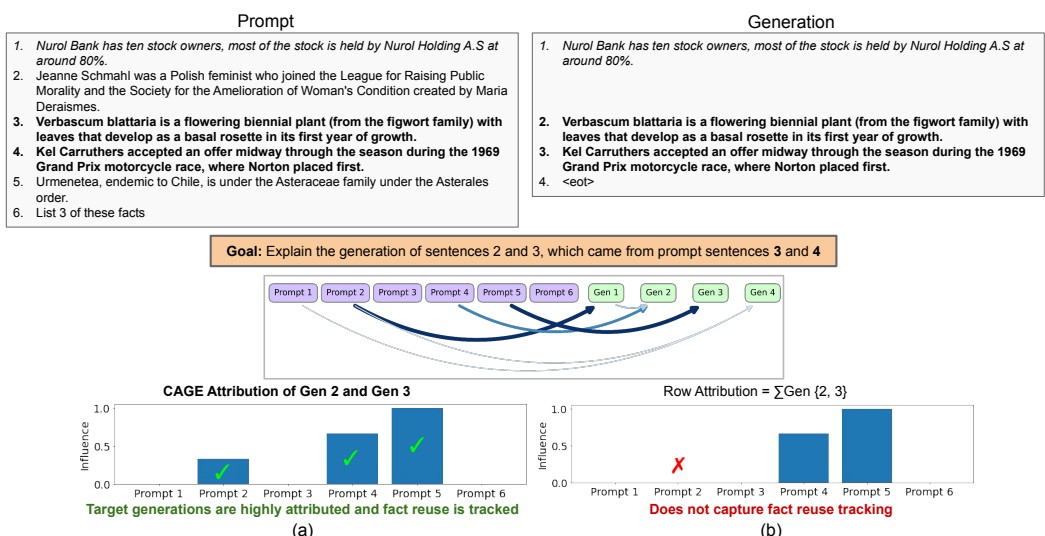

Figure 6: A Facts example with the Attn × IG base method on the Llama 3 8B model. CAGE demonstrates its ability to attribute the target generations and the model's reuse tracking behavior.

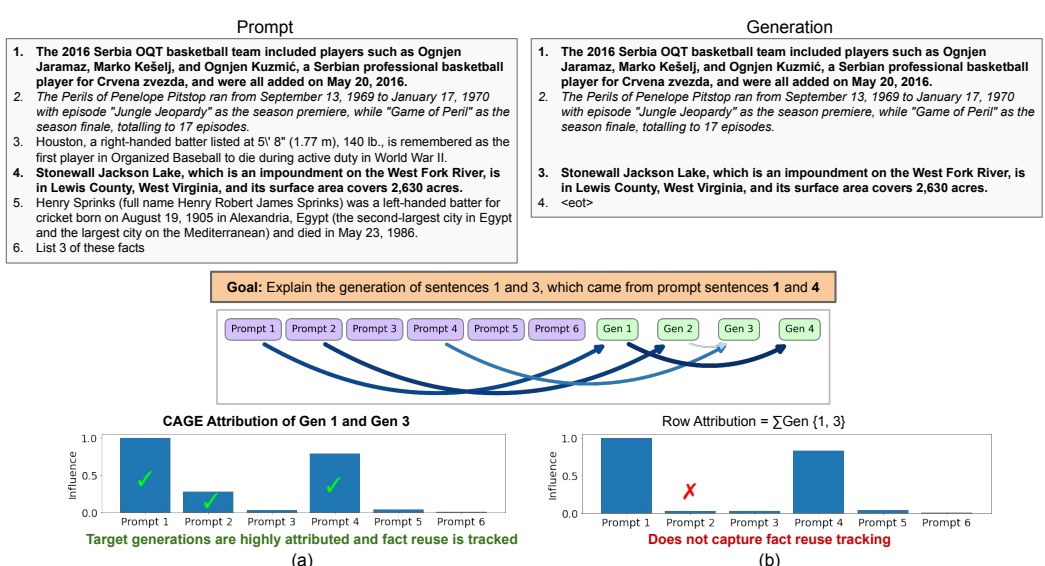

Figure 7: A Facts example with the perturbation base method on the Qwen 3 8B model. CAGE demonstrates its ability to attribute the target generations and the model's reuse tracking behavior.

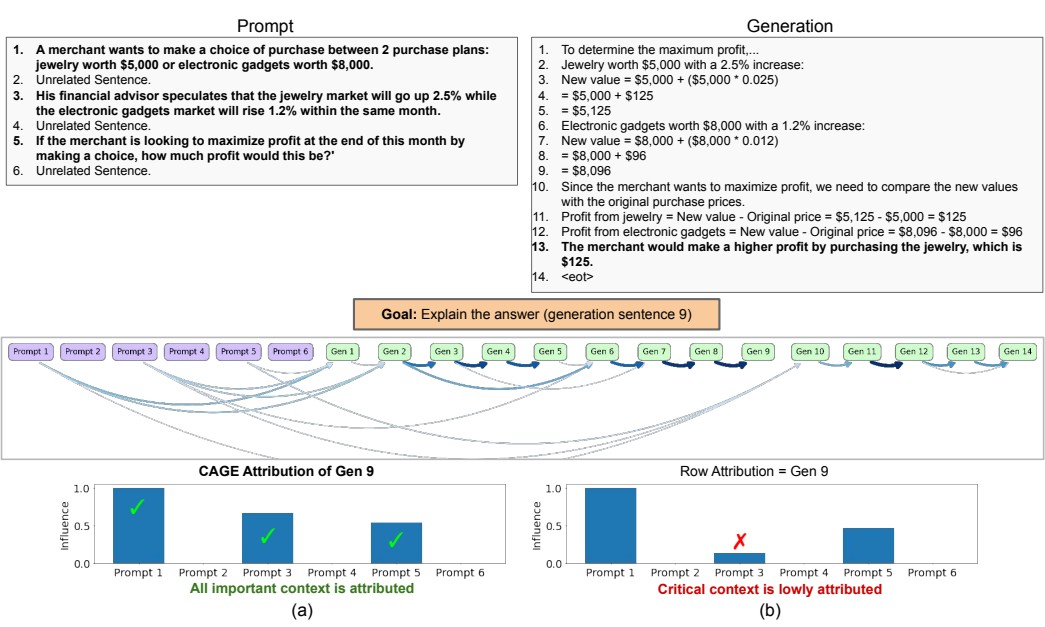

Figure 8: A Math example with the perturbation base method on the Llama 3 8B model. CAGE demonstrates its ability to properly attribute important prompt sentences.

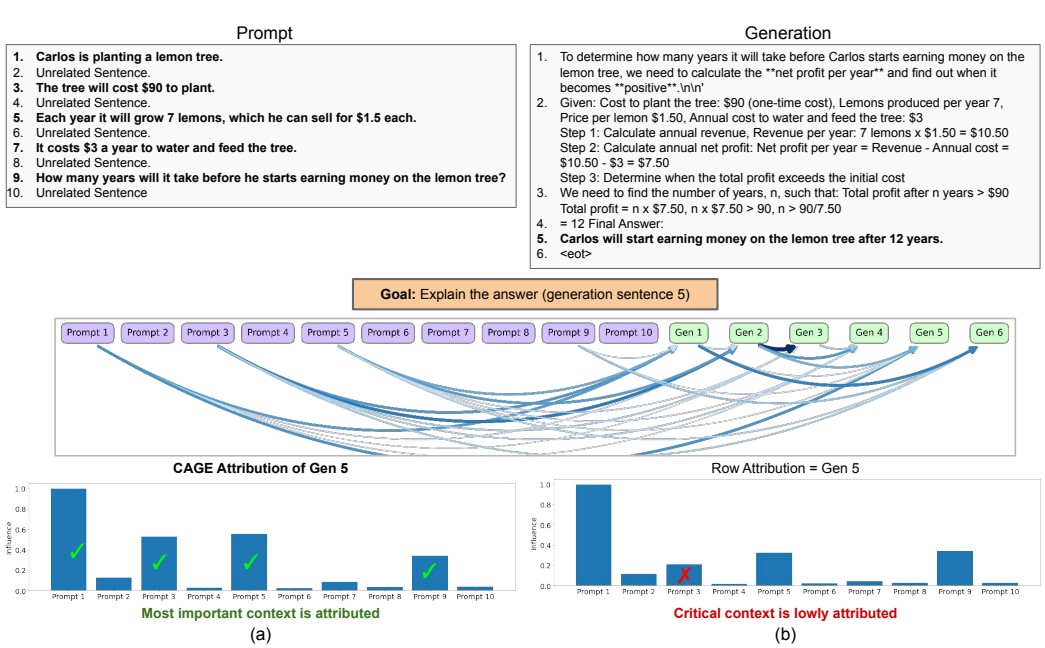

Figure 9: A Math example with the Attn × IG base method on the Qwen 3 8B model. CAGE demonstrates its ability to properly attribute important prompt sentences.

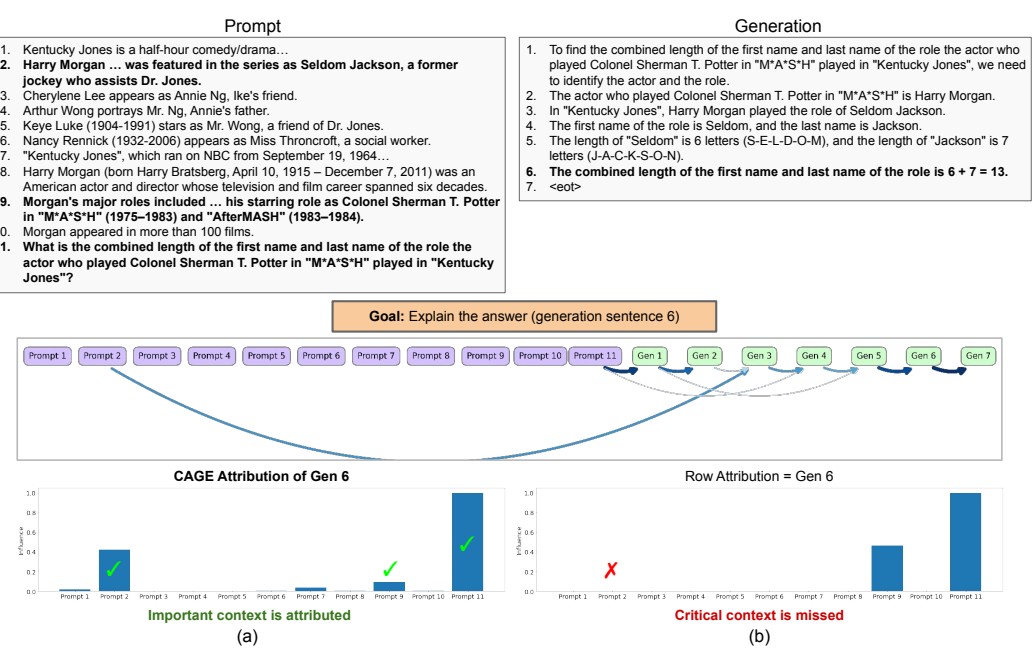

Figure 10: This figure shows a MorehopQA example with the perturbation base method on the Llama 3 8B model. We see in the prompt that sentences 2 and 9 contain relevant context and sentence 11 contains the question. Only CAGE attributes both sentences and the question, whereas row attribution ignores the first important prompt sentence altogether.

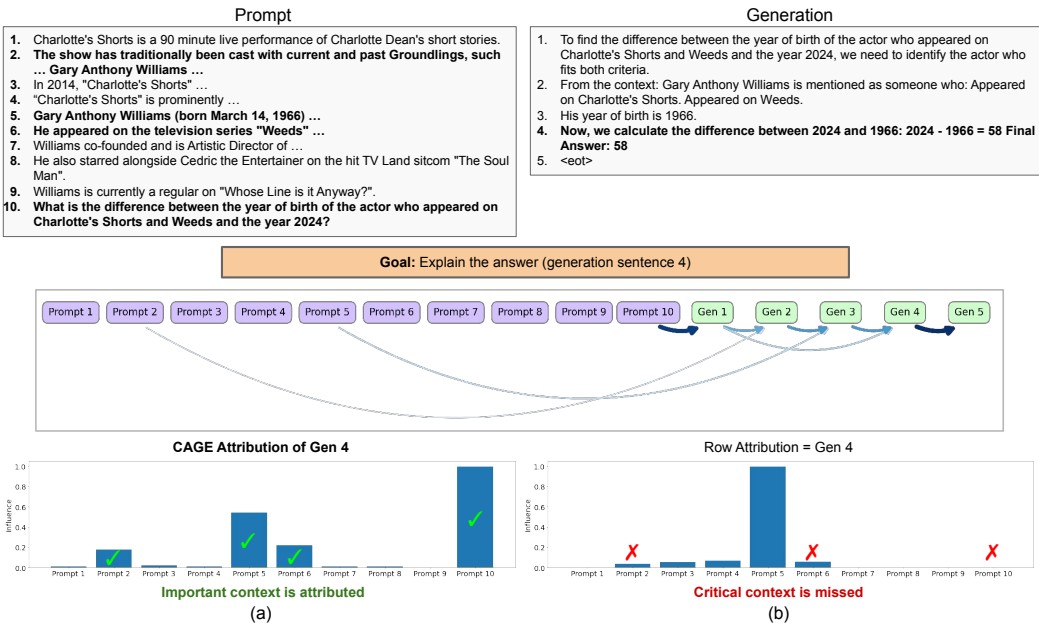

Figure 11: This figure shows a MorehopQA example with the perturbation base method on the Qwen 3 8B model. We see in the prompt that sentences 2, 5, and 6 contain relevant context and sentence 10 contains the question. Only CAGE attributes all three sentences and the question, whereas row attribution solely attributes sentence 5, which cannot be, by itself, responsible for generating the answer to the question.

## A.5 ABLATION STUDIES

In this section, we present quantitative and qualitative ablations of the CAGE framework. We evaluate three configurations: (1) the proposed CAGE, in which the adjacency matrix is non-negative and row-stochastic; (2) a variant that enforces non-negativity but removes row-normalization; and (3) a fully unnormalized variant using the raw attribution table (allowing negative values). In Tables 5 and 6 we run experiments over 100 inputs and report mean and standard deviation result for these configurations, which are denoted as "$\sum = 1$", "$\max(x, 0)$", and "None", respectively.

Table 5 reports input perturbation metrics for Qwen 3 8B on MorehopQA. Each column corresponds to one of the above configurations, allowing us to examine both (i) how CAGE improves each base attribution method and (ii) how the design choices affect CAGE itself. Across all base methods, all three configurations of CAGE outperform the corresponding base attributions, demonstrating that our influence propagation formulation is beneficial regardless of normalization. However, comparing the CAGE columns reveals that relaxing either non-negativity or row-normalization consistently degrades performance, confirming that the proposed formulation yields the most faithful attributions.

Table 5: Faithfulness evaluation via input perturbation metrics on MorehopQA Qwen 3 8B with normalization ablation

| Model and Dataset | | Qwen 3 8B - MorehopQA | | | | | |
|---|---|---|---|---|---|---|---|
| Normalization | | $\sum = 1$ | | $\max(x, 0)$ | | None | |
| Metric ($\downarrow$) | | RISE | MAS | RISE | MAS | RISE | MAS |
| Attn $\times$ IG | Row | $0.364 \pm 0.16$ | $0.464 \pm 0.18$ | $0.364 \pm 0.16$ | $0.464 \pm 0.18$ | $0.364 \pm 0.16$ | $0.463 \pm 0.18$ |
| | CAGE (ours) | $\mathbf{0.351} \pm 0.15$ | $\mathbf{0.444} \pm 0.16$ | $\mathbf{0.346} \pm 0.15$ | $\mathbf{0.441} \pm 0.16$ | $\mathbf{0.354} \pm 0.16$ | $\mathbf{0.451} \pm 0.17$ |
| Pert. | Row | $0.241 \pm 0.12$ | $0.360 \pm 0.19$ | $0.241 \pm 0.12$ | $0.360 \pm 0.19$ | $0.246 \pm 0.12$ | $0.363 \pm 0.17$ |
| | CAGE (ours) | $\mathbf{0.186} \pm 0.07$ | $\mathbf{0.250} \pm 0.11$ | $\mathbf{0.212} \pm 0.09$ | $\mathbf{0.325} \pm 0.16$ | $\mathbf{0.214} \pm 0.09$ | $\mathbf{0.332} \pm 0.16$ |
| Wins | | 2/2 | 2/2 | 2/2 | 2/2 | 2/2 | 2/2 |

Table 6: Faithfulness evaluation via input perturbation metrics on MorehopQA Qwen 3 14B with normalization ablation

| Model and Dataset | | Qwen 3 14B - MorehopQA | | | | | |
| --- | --- | --- | --- | --- | --- | --- | --- |
| Normalization | | $\sum = 1$ | | $\max(x,0)$ | | None | |
| Metric ($\downarrow$) | | RISE | MAS | RISE | MAS | RISE | MAS |
| Attn $\times$ IG | Row | $0.335 \pm 0.14$ | $0.438 \pm 0.15$ | $0.335 \pm 0.14$ | $0.438 \pm 0.15$ | $0.335 \pm 0.14$ | $0.438 \pm 0.14$ |
| | CAGE (ours) | $\mathbf{0.332} \pm 0.13$ | $\mathbf{0.433} \pm 0.12$ | $\mathbf{0.332} \pm 0.13$ | $\mathbf{0.433} \pm 0.12$ | $\mathbf{0.332} \pm 0.13$ | $\mathbf{0.433} \pm 0.12$ |
| Pert. | Row | $0.179 \pm 0.11$ | $0.253 \pm 0.16$ | $\mathbf{0.179} \pm 0.11$ | $0.253 \pm 0.16$ | $0.182 \pm 0.11$ | $0.271 \pm 0.16$ |
| | CAGE (ours) | $\mathbf{0.139} \pm 0.06$ | $\mathbf{0.179} \pm 0.07$ | $0.183 \pm 0.13$ | $\mathbf{0.205} \pm 0.13$ | $\mathbf{0.180} \pm 0.11$ | $\mathbf{0.254} \pm 0.13$ |
| Wins | | 2/2 | 2/2 | 1/2 | 2/2 | 2/2 | 2/2 |

Table 6 presents the same ablation for the larger Qwen 3 14B. The results mirror those of the 8B model. The proposed CAGE configuration again yields the strongest improvements, while the relaxed variants perform worse. This supports the robustness of our design choices and further shows that CAGE continues to provide substantial gains even on larger models beyond those in the main paper.

To illustrate these effects qualitatively, Figures 12 and 13 present an example from the Facts dataset. These figures demonstrate how removing row-stochasticity or non-negativity can fundamentally distort the outcome of the propagation process.

Figure 12 shows CAGE applied to a perturbation attribution when row stochasticity is removed. While (a) shows the expected behavior of CAGE under the proposed formulation, (b) shows that propagation via Eq. 4 produces exploding values, causing influence on the two target sentences to be overwhelmed by influence on the non-target sentence. As a result, only the non-target generated sentence appears influential. The row-stochastic formulation of CAGE explicitly prevents such explosions by inherently constraining values to be at most one, yielding more stable influence paths.

Figure 13 evaluates CAGE for an IG base method when both the non-negativity and row stochasticity constraints are removed. Again, (a) shows correct behavior for the proposed CAGE. In contrast, (b) reveals two failure modes. First, negative attributions on the diagonal, which reflect inhibitory effects, are propagated as negative coefficients, yielding recurrent sign flips and erasing influence on key prompt sentences as cumulative sums become negative. This contradicts the intended semantics of the mediated influence captured in the attribution table. Second, magnitudes again explode, leaving only the non-target generated sentence with a visibly positive attribution. Together, these artifacts show that permitting negative or unnormalized influence fundamentally destabilizes the proposed propagation mechanism.

Overall, the ablations demonstrate that the non-negative, row-stochastic formulation of CAGE is essential for producing stable, interpretable, and faithful influence graphs. Moreover, the results in Table 6 confirm that these benefits extend to larger model scales.

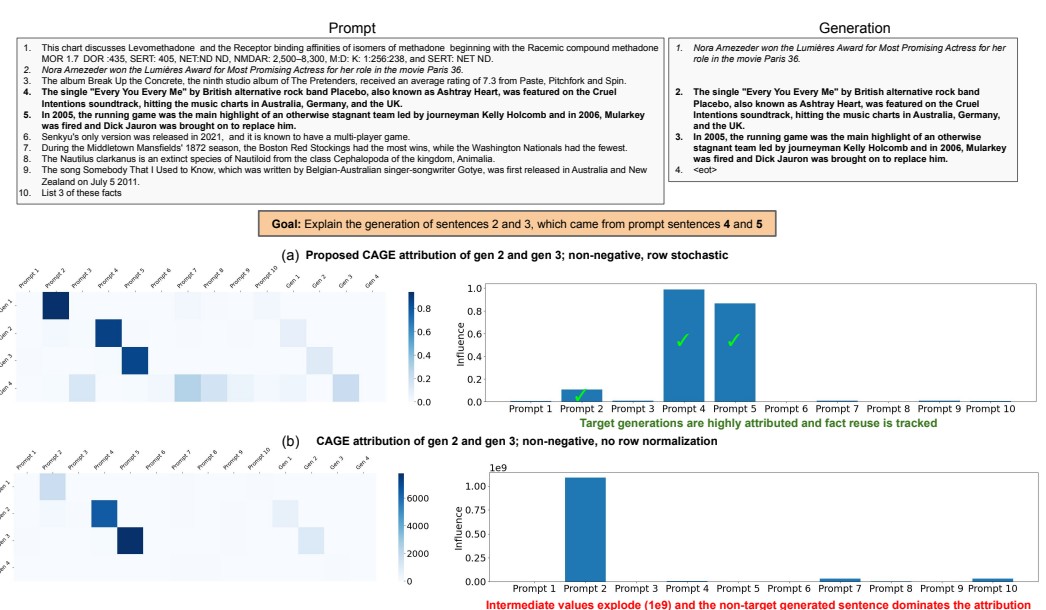

Figure 12: This figure illustrates an ablation test of the CAGE framework for a perturbation base method on the Facts dataset. In (a) CAGE, as proposed, properly attributes the input. In (b), when the row stochastic normalization is removed, value explosion occurs, leaving only the non-target generated sentence to appear important. The row stochastic formulation of CAGE is defined to avoid this problem.

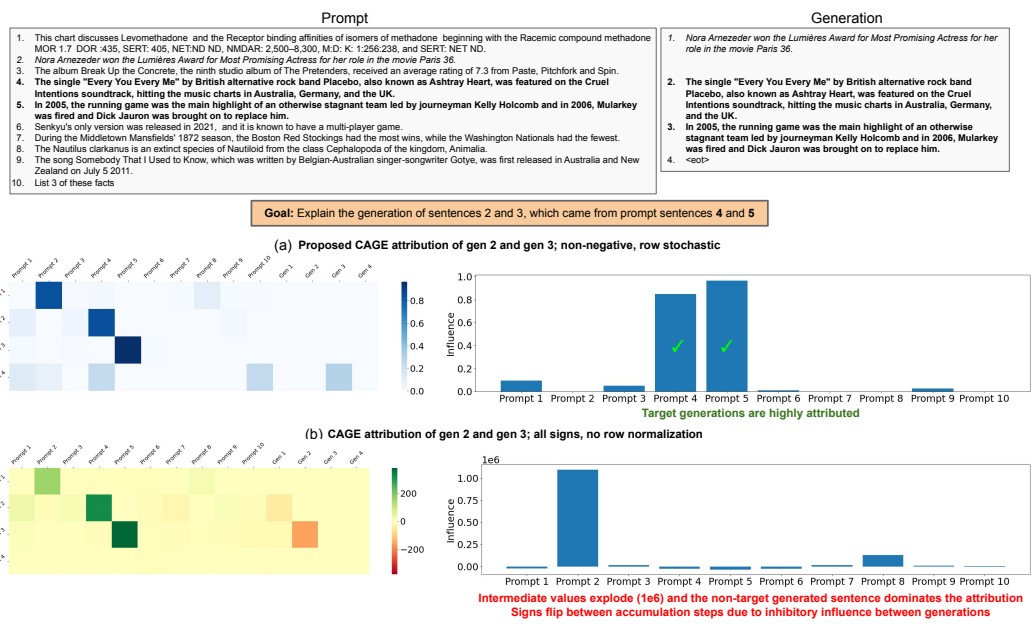

Figure 13: This figure illustrates an ablation test of the CAGE framework operating on the IG base method for the Facts dataset. In (a) CAGE, as proposed, properly attributes the input. In (b), when both non-negativity and row stochastic normalization are removed, value erasure due to negative signs and value explosion occur, leaving only the non-target generated sentence to be positively attributed. Hence, the non-negative, row stochastic formulation of CAGE are required to ensure proper function.

