# OpenReview forum: "Explaining the Reasoning of Large Language Models Using Attribution Graphs"
_ICLR.cc/2026/Conference — Submitted to ICLR 2026_

### Official Review · Reviewer_4PaL · 2025-10-27

**Soundness:** 2
**Presentation:** 3
**Contribution:** 2
**Rating:** 2
**Confidence:** 5

**Summary:**

The paper introduces CAGE, a context attribution method that builds a causal row stochastic graph over prompt and generated tokens and propagates influence to capture prompt to output and intergenerational effects in autoregressive language models. Experiments on facts, math, and multihop question answering with llama and qwen show consistent gains, with large faithfulness improvements and perfect wins in some settings.

**Strengths:**

* **Clear problem and fix:** Prior methods miss how earlier outputs affect later ones. CAGE builds a simple cause-and-effect graph that captures this and solves the gap directly.
* **General and clean:** It works on top of any attribution method, with straightforward normalization and a closed-form formula to spread influence, making it easy to reuse.
* **Strong results:** Consistent gains across models and datasets—up to 40% better on faithfulness, 17/20 wins on attribution coverage, and 40/40 wins in some tests.

**Weaknesses:**

* The paper enforces a row stochastic constraint (incoming edge weights sum to 1) without proper mathematical justification. This creates a critical logical flaw. The constraint assumes that 100% of a token's generation can be attributed to previous tokens, which contradicts how LLMs actually work. LLMs have inherent randomness, learned biases, and model parameters that contribute to generation beyond just the input context. By forcing weights to sum to 1, the method artificially inflates attribution scores when there are few contributing tokens, potentially creating misleading explanations

* The normalization in Equation (1) using Φ(x) = max(x, 0) can cause division by zero issues if all attribution scores for a token are negative. This is a serious mathematical flaw in the method.

* The mathematical formulation for computing context attribution has a fundamental error:
a^xτ = Aτ,: + Σ(i=1 to τ-1) Aτ,i · Ai, This assumes attribution influences combine linearly and additively, which is problematic because It treats intermediate tokens as simple linear transmitters of influence, ignoring the non-linear transformations in transformer architectures

* Evaluation is underpowered: no ablations relaxing causality or row-stochasticity to test necessity; reported gains lack confidence intervals or significance tests on the 500/250-example sets; key recent baselines (e.g., Barkan et al., EMNLP 2024) are omitted; and results exclude larger 14B–80B models—leaving generality and robustness unproven.

[1] Oren Barkan, Yonatan Toib, Yehonatan Elisha, Jonathan Weill, and Noam Koenigstein. Llm explainability via attributive masking learning. In Findings of the Association for Computational Linguistics: EMNLP 2024, pp. 9522–9537, 2024.

**Questions:**

Same as weakness

---

> ### Author Response · Authors · 2025-11-19
> **To Reviewer 4PaL**
>
> Thank you for your feedback. We appreciate your statements that the paper finds a valuable problem and provides a clear solution while being method agnostic and reusable, obtaining strong results in the process. Below are the answers to the questions unique to your review. Please see the general response for answers to questions that were common among reviewers.
>
> ---
>
> > Q1: The normalization in Equation (1) using $\phi(x)=\text{max}(x,0)$ can cause division by zero issues if all attribution scores for a token are negative. This is a serious mathematical flaw in the method.
>
> We appreciate the reviewer for pointing out this corner case. Indeed, if all attribution scores for a token are negative, the denominator in Eq. (1) could be zero after applying $\phi(x)=\text{max}(x,0)$. In practice, this is a very unlikely occurrence. Regardless, our code implementation does follow the standard convention of adding a small $\epsilon$ to the denominator to prevent division by zero.
>
> ---
>
> > Q2: Reported gains lack confidence intervals or significance tests on the 500/250-example sets; key recent baselines (e.g., Barkan et al., EMNLP 2024) are omitted; and results exclude larger 14B–80B models—leaving generality and robustness unproven.
>
> We agree that reporting confidence intervals is important for assessing evaluation robustness. In the revised version, we now include standard deviation values alongside the mean values for all tables. These values were computed during the original runs but were omitted from the initial draft for text readability.
>
> We appreciate the reviewer’s suggestion. In lines 40–41 of our paper, we had cited 2 papers from Barkan et al. (including this one) and acknowledge that the methods in this work evaluate autoregressive LLMs in a classification setting, where the model is prompted to produce a single output token. This is not the task that we wish to explain with this work, as a CAGE explanation for a single token generation is no different than existing explanation approaches because no multi-step influence propagation occurs (please see Section 4, paragraph 1, sentences 2 and 3 of the CAGE paper for this existing statement). Since CAGE specifically targets the explanation of multi-token autoregressive generation, the baselines in the cited work of Barkan et al. are not directly comparable.
>
> We agree that extending evaluation to larger models is valuable (models with more than 8B parameters). Autoregressive LLM attribution methods are, however, very expensive due to the need for at least one additional forward pass per generated token (after the initial generation) and, potentially, backward passes. Methods like IG which require numerous forward and backward passes are even slower. Due to this, and from what the authors have observed in the literature (Cohen-Want et al. 2025; Zhao et al., 2024; Cifka et al., 2022), we believe the 8B models we evaluate are representative of and comparable to existing work. We have run a small set of input perturbation tests for MorehopQA on Qwen3-14B that are added to the paper in Table 6 as part of the ablation studies. We see that on this larger model CAGE still creates significant improvements in faithfulness.
>
> - Benjamin Cohen-Wang, Yung-Sung Chuang, and Aleksander Madry. Learning to attribute with attention.
> - Zhixue Zhao and Boxuan Shan. Reagent: A model-agnostic feature attribution method for generative language models.
> - Ondrej Cífka and Antoine Liutkus. Black-box language model explanation by context length probing.

---

### Official Review · Reviewer_hQNq · 2025-10-29

**Soundness:** 2
**Presentation:** 2
**Contribution:** 2
**Rating:** 4
**Confidence:** 4

**Summary:**

CAGE builds an attribution graph over prompt and generated tokens (causal, row-stochastic), then marginalizes along paths to explain outputs; this improves attribution coverage and perturbation-based faithfulness versus row-summation baselines across Llama/Qwen on Facts, Math, and MorehopQA.

**Strengths:**

* Captures inter-generational influence (not just prompt→token), aligning with CoT behavior.

* Principled construction (nonnegative, row-stochastic adjacency; DAG; closed-form total influence)

* Consistent empirical gains (AC ↑ max/avg 134%/40%; faithfulness wins 40/40; up to 30%/11% improvement).

* Method-agnostic: wraps perturbation, CLP, IG, Attn×IG, ReAGent.

**Weaknesses:**

* Because CAGE applies Φ(x)=max(x,0) and then row-normalizes the attribution table into a stochastic adjacency, it discards inhibitory (negative) effects and collapses absolute magnitudes into relative shares that must sum to one, so negative influence and true effect size cannot be represented.

* The faithfulness tests remove entire prompt sentences and replace them with EOS tokens, which can introduce distribution shift and confound measured effects with artifacts of degraded input rather than true causal importance

* The method assumes that “influence” from earlier words simply adds up and passes straight through the sequence. Transformers don’t work that way, they apply non-linear, state-dependent updates where effects can interact or cancel.

**Questions:**

* Please add ablation studies to increase the faithfulness of the method.

* Also please refer to weakness as questions.

---

> ### Author Response · Authors · 2025-11-19
> **To Reviewer hQNq**
>
> Thank you for your feedback. We appreciate your recognition that the construction of our framework is aligned with CoT behavior, principled, method agnostic, and produces strong empirical gains. Below are the answers to the questions unique to your review. Please see the general response for answers to questions that were common among reviewers.
>
> ---
>
> > Q1: The faithfulness tests remove entire prompt sentences and replace them with EOS tokens, which can introduce distribution shift and confound measured effects with artifacts of degraded input rather than true causal importance.
>
> This is true, and this is a very well-known issue in the attribution literature. This choice of perturbation baseline is an open problem which many attribution methods and evaluation metrics have attempted to solve in numerous ways (Kapishnikov et al. 2019; Sturmfels et al. 2020; Haug et al., 2021). We agree with the reviewer that this process is imperfect, but these perturbation metrics are well-known, popular, and accepted, and our choice of removal by EOS token replacement follows the existing literature (Liu et al. 2024).
>
> - Andrei Kapishnikov, Tolga Bolukbasi, Fernanda Viégas, and Michael Terry. XRAI: Better Attributions Through Regions.
> - Pascal Sturmfels, Scott Lundberg, amd Su-In Lee. Visualizing the Impact of Feature Attribution Baselines.
> - Johannes Haug, Stefan Zürn, Peter El-Jiz, and Gjergji Kasneci. On Baselines for Local Feature Attributions.
> - Fengyuan Liu, Nikhil Kandpal, and Colin Raffel. AttriBoT: A Bag of Tricks for Efficiently Approximating Leave-One-Out Context Attribution.

---

### Official Review · Reviewer_mXB2 · 2025-10-30

**Soundness:** 3
**Presentation:** 3
**Contribution:** 2
**Rating:** 4
**Confidence:** 3

**Summary:**

This paper introduces Context Attribution via Graph Explanations (CAGE), a novel framework for explaining the reasoning of autoregressive Large Language Models (LLMs). The authors argue that existing "row attribution" methods provide incomplete explanations because they only measure the direct influence of prompt tokens on each generated token, ignoring the crucial influence of intermediate generations. The CAGE framework addresses this limitation by constructing an attribution graph, a directed acyclic graph that models the causal flow of influence throughout the entire generation process. To generate a final context attribution for a specific output, CAGE marginalizes the influence of intermediate tokens by tracing all causal paths from the output back to the prompt tokens.

**Strengths:**

- Current "row attribution" methods treat each generated token's relationship to the prompt in isolation, which is fundamentally incompatible with the sequential, stateful nature of autoregressive generation, especially in chain-of-thought processes. The proposed CAGE framework is a novel and principled solution to this problem, utilizing a causal perspective on the generation process.
- The paper is well-written and easy to follow. The motivation is clearly laid out in the introduction, and the distinction between CAGE and prior work is made both in the text and through effective visualizations. Especially, Figure 3 greatly aids in understanding the influence propagation process.
- By providing a tool to trace the flow of information through intermediate generation steps, CAGE claims to offer a more faithful and complete picture of a model's reasoning process than was previously possible with attribution methods. The framework is general and can be applied on top of any existing or future base attribution method.

**Weaknesses:**

- The model of influence propagation, calculated via matrix multiplication ($A_{\tau,:}^{\prime}=A_{\tau,:}+A_{\tau,\tau-1}\cdot A_{\tau-1,:}$), implicitly assumes that influence propagates through the network in a way that can be modeled by a linear combination of path weights. The paper could be strengthened by acknowledging this as a simplifying assumption and briefly discussing why it is a reasonable one in this context, or contemplating what might be lost by this linearization of the influence flow.

- The construction of the adjacency matrix $A$ involves clipping all attribution scores to be non-negative ($\Phi(x) = \max(x, 0)$) before row-normalization. Many attribution methods can produce meaningful negative scores, indicating that a feature actively suppresses a certain output. By discarding this information, the model may lose a crucial part of the explanation. For example, in the "Facts" dataset, a model might rely on already-generated facts to down-weight the probability of generating them again. This "negative" influence is an important part of the reasoning process that is currently ignored. The paper would benefit from a discussion of this design choice and its potential limitations.

- The context attribution evaluation relies heavily on the Attribution Coverage (AC) metric, which is newly introduced in this paper. While novel metrics can be valuable, their introduction requires strong justification. The core assumption of the AC metric—that attribution should be uniformly distributed across all ground-truth sentences—is not well-defended and may not be appropriate for complex reasoning tasks. It is highly plausible that certain pieces of information are more pivotal than others. By penalizing non-uniform distributions, the AC metric may unfairly disadvantage methods that correctly identify and focus on the most crucial context. The paper would be more convincing if it either provided a stronger theoretical justification for the uniformity assumption or supplemented the AC results with a more standard metric.

**Questions:**

- The AC metric, as defined in the paper, rewards explanations where attribution is spread uniformly across all ground-truth sentences. However, in complex reasoning tasks, it's plausible that some pieces of evidence are more critical than others. Why is uniform attribution considered an ideal property, rather than allowing for a non-uniform distribution that might more accurately reflect the model's focus? Furthermore, could the authors provide the rationale for selecting the specific range $[\frac{1}{2}\mathbb{E}(a_{GT}), \frac{3}{2}\mathbb{E}(a_{GT})]$ for this metric?
- To isolate the goal of identifying all relevant prompt sentences, have the authors considered an alternative metric that measures coverage without enforcing a uniformity constraint? For example, a metric based on the average rank of ground-truth sentences (when all prompt sentences are sorted by their attribution scores) could provide a more direct assessment of whether an explanation method successfully highlights the most important information.
- The faithfulness evaluation in line 463 mentions calculating the "area under this perturbation curve". Could the authors explicitly define the axes of this curve? Based on the procedure described, is it correct to assume the x-axis represents the fraction of prompt sentences removed and the y-axis represents the probability of generating the original output?

---

> ### Author Response · Authors · 2025-11-19
> **To Reviewer mXB2**
>
> Thank you for your feedback. We appreciate you highlighting that we solve a critical issue present with existing AR LLM attribution methods and that CAGE presents a more faithful attributions while being base method agnostic. We also appreciate the complements for the writing, organization, and visualizations. Below are the answers to the questions unique to your review. Please see the general response for answers to questions that were common among reviewers.
>
> ---
>
> > Q1: To isolate the goal of identifying all relevant prompt sentences, have the authors considered an alternative metric that measures coverage without enforcing a uniformity constraint? For example, a metric based on the average rank of ground-truth sentences (when all prompt sentences are sorted by their attribution scores) could provide a more direct assessment of whether an explanation method successfully highlights the most important information.
>
> We thank the reviewer for their suggestion of a different metric. If we understand correctly, the reviewer is suggesting an internal consistency metric that evaluates how consistently an attribution ranks the important sentences for different inputs. The authors believe this could be a valuable metric if the ranked importance of the input sentences was known, but we do not know if it is a good fit in this context. We ask the reviewer to please clarify or add more context to improve the authors' understanding of the suggestion. We additionally note that the input perturbation tests employed do perturb in order of most attributed sentence and directly measure the importance of each sentence to the model through this process. This is, in a way, a proxy to the suggested method that does not need ground truth importance values for each sentence in the input.
>
> ---
>
> > Q2: The faithfulness evaluation in line 463 mentions calculating the "area under this perturbation curve". Could the authors explicitly define the axes of this curve? Based on the procedure described, is it correct to assume the x-axis represents the fraction of prompt sentences removed and the y-axis represents the probability of generating the original output?
>
> Yes, this assumption is correct. The perturbation tests employed here are common in the literature, which led to our choice to exclude some details of their definition. The multi-step perturbation process iteratively (and cumulatively) ablates the input, measuring the confidence of the prediction at each step. The perturbation curve formed is then a generation probability vs perturbation percentage curve as the reviewer says. With the extra page for the camera-ready version, we can include this small detail to enhance a reader's understanding. Please see the first paragraph of 4.5 for this change.

---

### Official Review · Reviewer_kaqK · 2025-11-03

**Soundness:** 3
**Presentation:** 3
**Contribution:** 2
**Rating:** 6
**Confidence:** 4

**Summary:**

This paper proposes CAGE (Context Attribution via Graph Explanations), a framework for explaining the reasoning process of autoregressive large language models (LLMs) through attribution graphs. The authors argue that existing row-wise attribution methods fail to effectively capture inter-generational dependencies during token generation, resulting in incomplete and inaccurate explanations. CAGE addresses this issue by constructing a directed acyclic graph (DAG) that preserves both causality and row stochasticity, and by marginalizing intermediate contributions along graph paths to compute contextual attributions. Experiments across multiple models and datasets demonstrate that CAGE achieves significant improvements in attribution faithfulness and coverage.

**Strengths:**

1. Clear motivation and presentation. The paper articulates its motivation clearly and presents the research problem and solution in a well-structured and intuitive manner.

2. The empirical evaluations cover four models, three different task datasets, five base attribution methods, and multiple evaluation metrics. CAGE achieves the best performance in 85% of comparisons (17/20 on the AC metric, 40/40 on faithfulness), lending strong empirical support to its claims.

**Weaknesses:**

1. Limited technical novelty and theoretical depth. While the paper identifies clear shortcomings in row-wise attribution methods, the proposed CAGE framework primarily relies on standard graph-based computations (e.g., path accumulation operations). The approach appears straightforward. The authors could further clarify the technical or theoretical challenges involved in the research, and articulate any new conceptual insights it provides regarding the semantics of contextual attribution.

2. Potential issues with the AC metric design. The AC metric assumes that ground-truth sentence attributions follow an approximately uniform distribution. However, in reality, different sentences naturally contribute unequally to overall generation. This assumption may unintentionally penalize explanations that accurately reflect such differences in importance.

**Questions:**

1. What is the main challenge and novelty of the CAGE framework?

---

> ### Author Response · Authors · 2025-11-19
> **To Reviewer kaqK**
>
> Thank you for your feedback. We appreciate your supporting score and your identification that the motivation and presentation of the paper is clear, well-structured, and intuitive as well as recognizing the strong empirical performance of CAGE. Below are the answers to the questions unique to your review. Please see the general response for answers to questions that were common among reviewers.
>
> ---
>
> > Q1: The work is technically limited in that it uses existing graph-based computations and appears straightforward. What is the main challenge and novelty of the CAGE framework?
>
> CAGE presents a new problem formulation and a principled solution for AR LLM context attribution. Prior work treats the generation as a single-step prediction, which is a critical flaw. Our key novelty is identifying and formalizing the multi-step inter-generational influence that row attributions fail to capture. The attribution graph that is used to reach our solution does employ familiar components and algorithms, but it is a new abstraction for explanation that (a) introduces a causal, inter-generation influence structure, (b) enables closed-form computation of multi-step influence contributions, and (c) yields an interpretable visualization that exposes reasoning chains that existing approaches cannot. CAGE is not simply a combination of or improvement on existing methods; it is a novel framework that rethinks the attribution problem for AR LLMs and presents a new class of explanations to achieve consistent improvements across multiple models, datasets, and base methods.
>
> We have updated the conclusion to reinforce the novelty.

---

### Author Response · Authors · 2025-11-19
**Message to All Reviewers, Part 1**

We thank all reviewers for their valuable feedback. Reviewers [mXB2], [hQNq], and [4PaL] state that CAGE provides a novel solution to an obvious flaw in autoregressive LLM explainability and appreciated that CAGE provides a method-agnostic solution. [kaqK], [hQNq], and [4PaL] additionally acknowledge the significant improvements in experimental evaluation achieved by CAGE. [kaqK] and [mXB2] found the writing and presentation of the paper to be clear and effective.

We would like to reiterate that CAGE's main contributions are twofold:
1. This is the first work to identify a critical issue in current autoregressive (AR) LLM attribution approaches: inter-generational influence is not captured or integrated to build context attributions. By identifying and solving this issue, we open research for further improvements to attribution methods for AR LLMs.
2. We define a foundational solution to this new problem that relies on an intuitive and novel graph construction. This not only results in improved quantitative evaluations but is itself a unique abstraction and interpretability visualization for the LLM.

Reviewer concerns primarily focus on methodological design choices. Specifically, they question the assumptions and relaxations used in the non-negative, row-stochastic graph formulation and in the linear propagation construction of the CAGE context attribution. Some reviewers also raised concerns regarding the design of the attribution coverage metric. Below, we address these points in detail and clarify how similar assumptions are commonly employed in related work and are well supported in the historical literature on black-box model explanation. Overall, we believe that identifying and solving a new problem that can shape the future direction of LLM explanation research represents a meaningful contribution to the field. We further believe that the significance of this contribution outweighs the methodological concerns, particularly since our design choices are consistent with those in well-cited prior work. Ultimately, however, we recognize that this assessment rests with the reviewers.

For this initial rebuttal, we have made changes to the main document which are denoted by blue text. These include additions to the main body text and the Appendix in accordance with questions and suggestions from the reviewers. Below is a response to questions which were common among reviewers and each reviewer has their unique questions answered in individual replies.

---

> ### Author Response · Authors · 2025-11-19
> **Message to All Reviewers, Part 2**
>
> >Q1: Reviewers [mXB2], [hQNq], [4PaL] question the construction of the attribution graph. Mainly, they hold issue with the use of $\phi(x)=\text{max}(x,0)$ and row-normalization within the attribution table because it (a) discards negative influence and (b) assumes 100% of a token's generation can be attributed to previous tokens while artificially inflating influence. Reviewers [hQNq] and [4PaL] ask for ablations relaxing the causality and row stochasticity properties of the design.
>
> We thank the reviewers for highlighting important aspects of the attribution graph construction. Our use of $\phi(x)=\text{max}(x,0)$ and row normalization is a deliberate modeling choice motivated by considerations that are supported by the existing literature.
> - The authors recognize that negative values captured by attributions often reflect inhibitory effects, which are valuable for understanding behavior, but propagating these values multiplicatively through this graph construction leads to unstable magnitudes, information erasure, and hard-to-interpret sign changes. This has led to many works in the past such as LRP (LRP - Binder et al., 2016) and GBP (Springenberg et al., 2014) to similarly discard intermediate negative values.
> - Our use of row normalization does make inherent simplifying assumptions about the transformer architecture, but the assumptions made are again aligned with the existing literature. In Rollout (Abnar and Zuidema, 2020), the formulation expects a conservation of information propagation through each layer of the network, relying on the row stochasticity of attention weights, and in LRP, they make a simplifying assumption that all information entering a layer of a network will exit, ignoring nonlinearities in the process. The use of the normalization abstracts away the model implementation, allowing for an effective explanation of high-level behavior and reasoning.
> - Overall, our CAGE framework treats the attribution table as defining a directed influence graph over generated tokens. To allow marginalization of influence along its paths, this graph must satisfy basic non-negativity and conservation constraints that are backed by the literature mentioned above. Enforcing these constraints ensures that the directional causality of the graph is meaningful and that the closed-form marginalization proposed in Eq. 4 is well defined.
>
> In response, we have added Section 3.4 to the paper to include this meaningful discussion. We have also added ablation studies on pages 19 through 21 including Tables 5 and 6 as well as Figures 12 and 13. These ablation studies show that both row normalization and non-negativity are critical parts of our construction. In future work, we agree that promising directions include changing the construction and influence propagation method to allow signed graph edges and relaxing the row stochasticity constraint as acknowledged in a new paragraph added to the reworked Conclusion and Future Work (Section 5).
>
> ---
>
> > Q2: Reviewers [mXB2], [hQNq], and [4PaL] discuss how our construction assumes influence propagates between tokens, through the network, in a linear manner with our definition: $\alpha^{x_\tau} = A_{\tau,:} + \sum_{i = \tau - 1}^{1}A_{\tau, i} \cdot A_{i, :}$. They state that transformers do not work this way.
>
> We thank the reviewers for this critique. We agree that the transformer model is highly nonlinear and that our problem formulation makes a linear assumption about how influence propagates through the network. The goal of this work was not to create a mechanistic understanding of LLMs, but instead to provide a foundational template that (a) exposes the failure modes of existing, incomplete row attribution methods and (b) better predicts influential inputs through its graph construction. This linear assumption serves this purpose. While there is room for improvement in future works that relax this constraining assumption, we believe this assumption is valid in the scope of the literature as many approaches for explanation such as local linear surrogate models (LIME - Ribeiro et al., 2016), first-order/Taylor influence methods (GBP - Springenberg et al., 2014; IG - Sundararajan et al., 2017), attention-flow and rollout (Abnar and Zuidema, 2020), and relevance-propagation frameworks (LRP - Binder et al., 2016) make equivalent assumptions. This discussion to clarify the assumptions we make, their motivation, and their justification is also included in the newly added Section 3.4.

---

> ### Author Response · Authors · 2025-11-19
> **Message to All Reviewers, Part 3**
>
> > Q3: Reviewers [kaqK] and [mXB2] question the design of the attribution coverage (AC) metric employed as part of the experimental evaluation. They note that the metric assumes sentence attributions will conform to a near-uniform distribution over important sentences. They ask: Why is uniform attribution considered an ideal property, rather than allowing for a non-uniform distribution that might more accurately reflect the model's focus? Furthermore, could the authors provide the rationale for selecting the specific range $[\frac{1}{2}\mathbb{E}(\alpha_{GT}), \frac{3}{2}\mathbb{E}(\alpha_{GT})]$ for this metric?
>
> We thank the reviewers for raising this valid concern. The authors agree that it is unrealistic to expect every important sentence to recieve the same portion of attribution mass. Thus, we relaxed the attribution coverage metric to accept a range of attribution mass on each important sentence based on how many there are. This is why we chose the acceptance range $[\frac{1}{2}\mathbb{E}(\alpha_{GT}), \frac{3}{2}\mathbb{E}(\alpha_{GT})]$. For a simple example, consider an input $[a, b, c]$ where $a$ and $c$ are important and $b$ is a distractor. The acceptable range would then be $[0.25, 0.75]$. Then, all of the following attributions of this input would get a score of $1$: $[0.25, 0, 0.75]$, $[0.5, 0, 0.5]$, $[0.75, 0, 0.25]$, and $[0.33, 0.33, 0.33]$.
>
> Additionally, the math dataset we employed is constructed such that every important sentence in the input (the non-distractors) have required context for answering the question, i.e. the question could not be answered with any one of the known important sentences missing. We recognize that the design of this metric is heuristic and imperfect, but due to our employed relaxation and the nature of the data over which we evaluate, we believe this is a satisfactory metric that is better included than not.
>
> We welcome suggestions for alternative evaluation metrics that serve as an equivalent ground truth test to this AC metric. If feasible within the remaining review window, we would do our best to implement such a metric and provide results. To reviewer [mXB2]: we acknowledge the metric you suggested, please see our direct reply.

---

### Author Response · Authors · 2025-12-01
**Message to AC After Changes to Review Process**

The initial reviews highlighted the novelty and relevance of the CAGE framework and recognized the value it brings to advancing the state of the art in LLM explanation. The primary reviewer concern regarded design choices in the methodology that abstract away model structure. In response, we provided a detailed discussion that explains how these choices follow established practices in the literature to enable explanation of complex models, and we added additional targeted ablation studies and evaluations to empirically support these choices. Furthermore, we have provided clarifying discussions and additional detail in the updated PDF in response to reviewer questions. We believe the concerns raised by the reviewers have been substantively resolved, and it's likely that the reviewers would have adjusted their scores if the discussion period continued. With these clarifications and additional results, we believe the submission meets the standards of novelty, rigor, and clarity expected for ICLR.

---

### Meta-Review · Area_Chair_zYos · 2026-01-07

**Summary:**

The paper identifies a critical limitation in existing row attribution methods, where they fail to capture inter-generational influence during token generation, treating each generated token's relationship to the prompt in isolation. To this end, the authors introduces CAGE, a framework for explaining autoregressive LLM reasoning through attribution graphs by constructing a directed acyclic graph that models causal influence flow throughout the generation process. The reviewers recognized that the observation that existing attribution methods ignore how earlier generated tokens influence later ones is both valid and well-motivated, particularly for chain-of-thought reasoning scenarios. While the reviewers appreciated that CAGE provides a method-agnostic solution that can wrap any existing attribution approach, their primary concerns were centered on methodological design choices in the attribution graph construction. Multiple reviewers questioned the use of non-negative clipping and row normalization, arguing that discarding negative attribution scores loses meaningful inhibitory effects and that forcing weights to sum to one artificially inflates influence when few tokens contribute.

To this end, the authors mentioned that their code implementation does follow the standard convention of adding a small noise to the denominator to prevent division by zero. However, looking at the normalization code, didn't show any such addition.

```    # normalize the max of a vector to 1
    def normalize_max(self, attribution_vector) -> torch.Tensor:
        if attribution_vector.max() > 0:
            attribution_vector = attribution_vector / attribution_vector.max()
        elif attribution_vector.max() <= 0:
            attribution_vector = - attribution_vector / attribution_vector.min()

        return attribution_vector
```

**Reviewer Concerns:**

While reviewers challenged i) the linear assumption about influence propagation, noting that transformers apply nonlinear, state-dependent updates and ii) the Attribution Coverage metric for its uniformity assumption, which may penalize explanations that correctly identify varying levels of importance among input sentences, the authors made genuine efforts to address these concerns (new ablation studies and detailed text) and note that similar assumptions are standard in the attribution literature.

The concerns on the methodological assumptions still remain unaddressed.

**Reviewer Scores:**

None of the reviewers engaged in discussions with the authors.

---

### Decision · Program_Chairs · 2026-01-26

Reject